

# Sensitivity of black carbon concentrations and climate impact to aging and scavenging

Marianne T. Lund[1,*], Terje K. Berntsen[1,2], Bjørn H. Samset[1]

*1 CICERO - Center for International Climate and Environmental Research, Oslo, Norway*

*2 Department of Geosciences, University of Oslo, Oslo, Norway*

*Corresponding author, m.t.lund@cicero.oslo.no, Phone: +47 22 85 86 94*

## Abstract

Despite recent improvements, significant uncertainties in global modeling of black carbon (BC) aerosols persist, posing important challenges for the design and evaluation of effective climate mitigation strategies targeted at BC emission reductions. Here we investigate the sensitivity of BC concentrations in the chemistry-transport model OsloCTM2 with the microphysical aerosol parameterization M7 (OsloCTM2-M7) to parameters controlling aerosol aging and scavenging. We focus on Arctic surface concentrations and remote region BC vertical profiles, and introduce a novel treatment of condensation of nitric acid on BC.

The OsloCTM2-M7 underestimates annual averaged BC surface concentrations, with a mean normalized bias of -0.55. The seasonal cycle and magnitude of Arctic BC surface concentrations is improved compared to previous OsloCTM2 studies, but model-measurement discrepancies during spring remain. High-altitude BC over the Pacific is overestimated compared with measurements from the HIPPO campaigns. We find that a shorter global BC lifetime improves the agreement with HIPPO, in line with other recent studies. Several processes can achieve this, including allowing for convective scavenging of hydrophobic BC and reducing the amount of soluble material required for aging. Simultaneously, the concentrations in the Arctic are reduced, resulting in poorer agreement with measurements in part of the region.

A first step towards inclusion of aging by nitrate in OsloCTM2-M7 is made by allowing for condensation of nitric acid on BC. This results in a faster aging and reduced lifetime, and in turn to a better agreement with the HIPPO measurements. On the other hand, model-





measurement discrepancies in the Arctic are exacerbated. Work to further improve this
parameterization is needed.
The impact on global mean radiative forcing (RF) and surface temperature response (TS) in our
experiments is estimated. Compared to the baseline, decreases in global mean direct RF on the
order of 10-30% of the total pre-industrial to present BC direct RF is estimated for the
experiments that result in the largest changes in BC concentrations.

We show that globally tuning parameters related to BC aging and scavenging can improve the
representation of BC vertical profiles in the OsloCTM2-M7 compared with observations. Our
results also show that such improvements can result from changes in several processes and often
depend on assumptions about uncertain parameters such as the BC ice nucleating efficiency
and the change in hygroscopicity with aging. It is also important to be aware of potential
tradeoffs in model performance between different regions. Other important sources of
uncertainty, particularly for Arctic BC, such as model resolution has not been investigated here.
Our results underline the importance of more observations and experimental data to improve
process understanding and thus further constrain models.


## 1 Introduction

Black carbon (BC) aerosols play an important role in the climate system through several
mechanisms including direct absorption of solar radiation (Bond et al., 2013; Myhre et al.,
2013), changing surface albedo (Flanner et al., 2009; Warren & Wiscombe, 1980), modification
of cloud properties and thermal stability (Koch & Del Genio, 2010; Lohmann & Feichter, 2005),
and influence on precipitation and circulation (Bollasina et al., 2014; Wang et al., 2015). The
potentially strong climate warming, combined with short atmospheric residence time and
harmful health impacts (Anenberg et al., 2012; Aunan et al., 2006; Shindell et al., 2011), has
made BC reductions an attractive mitigation measure (AMAP, 2015; Bowerman et al., 2013;
EPA, 2012; Grieshop et al., 2009; Kopp & Mauzerall, 2010; UNEP/WMO, 2011).

However, accurately representing the distribution of BC concentrations in global atmospheric
models remains challenging and considerable inter-model variability and model-measurement
discrepancies exist. In particular two features have been pointed out in several studies:





underestimation of the magnitude and difficulty capturing the seasonal cycle of Arctic BC
surface concentrations (e.g., (Eckhardt et al., 2015; Shindell et al., 2008)) and an overestimation
of high altitude BC concentrations over remote regions (e.g., (Koch et al., 2009b; Lee et al.,
2013; Samset et al., 2014; Schwarz et al., 2013; Wang et al., 2014)). Because the impact of the
aerosols on radiation and temperature depends strongly on altitude, such discrepancies lead to
uncertainties in the net climate impact of BC. While overestimating high altitude BC
concentrations can result in an overestimation of the BC radiative forcing (Samset & Myhre,
2011), too low surface concentrations may lead to an underestimation of the temperature
response due to the reduced efficacy of BC forcing with altitude (Ban-Weiss et al., 2011;
Flanner, 2013; Samset & Myhre, 2015). This in turn poses significant challenges for the design
and evaluation of effective BC mitigation strategies.

Several studies have explored how scavenging processes and uncertainties in emissions
contribute to the inter-model and model-measurement discrepancies. Some have investigated
how these processes influence transport of BC to the Arctic (Bourgeois & Bey, 2011; Browse
et al., 2012; Liu et al., 2011), others have focused on the vertical BC distribution in remote
regions (Fan et al., 2012; Hodnebrog et al., 2014; Kipling et al., 2016; Kipling et al., 2013).
In this study we use the chemical transport model OsloCTM2 (Sovde et al., 2008) with the
microphysical aerosol parameterization M7 (Vignati et al., 2004) (hereafter OsloCTM2-M7) to
investigate the sensitivity of modeled BC concentrations to changes in a range of parameters
related to aging and scavenging processes and how these influence the model-measurement
discrepancies, focusing simultaneously on Arctic surface concentrations and remote region
vertical distributions of BC.
The OsloCTM2 has been used in several multi-model studies of aerosol impacts (Balkanski et
al., 2010; Myhre et al., 2013; Schulz et al., 2006; Shindell et al., 2013; Textor et al., 2007).
These studies used a simplified bulk parameterization of carbonaceous aerosols. Lund and
Berntsen (2012) compared the M7 to this bulk parameterization and found improved
representation of modeled BC seasonal cycle and magnitude at high latitudes. However, a
comparison against BC vertical profiles measured during one aircraft campaign suggested that
M7 exacerbated high-altitude overestimation of concentrations.
Here we use updated inventories of anthropogenic and biomass burning emissions and three
years of model results, and further evaluate the OsloCTM2-M7 against a range of observations





from surface stations, flight campaigns, and snow samples. Next, we perform sensitivity
experiments to quantify the impact of changes in a range of physical and microphysical
parameters on the BC distribution. Our sensitivity experiments include a first step towards
accounting for gas-phase nitric acid condensation in the BC aging parameterization.
Measurements have shown that nitrate is frequently present in internal aerosol mixtures and
contribute to the aging of BC (Pratt & Prather, 2010; Shiraiwa et al., 2007), a process currently
excluded in many models. This process may also become more important in the future
following strong projected decreases in $SO_2$ emissions and increasing NOx and greenhouse gas
emissions (Bauer et al., 2007; Bellouin et al., 2011; Makkonen et al., 2012). Finally, we estimate
the subsequent impact of BC concentrations changes on global radiative forcing and surface
temperature response.
Section 2 describes the model setup and experiments, Sect. 3 presents and discusses results and
Sect. 4 gives the conclusions.

**2 Methodology**

*2.1 The OsloCTM2-M7*

The OsloCTM2 is a global off-line 3-dimensional chemistry transport model with transport
driven by meteorological data generated by the Integrated Forecast System (IFS) model at the
European Center for Medium Range Weather Forecast (ECMWF) (Sovde et al., 2008). The
model is run for 2008-2010 with a T42 resolution (approximately 2.8° x 2.8°) and 60 vertical
layers from the surface to 0.1 hPa.

The microphysical aerosol module M7 (Lund & Berntsen, 2012; Vignati et al., 2004) includes
the main aerosol species sea salt, mineral dust, sulfate and organic carbon, in addition to BC.
Aerosols are represented by seven modes with size distribution given by a lognormal
distribution function. BC aerosols are separated into soluble (mixed) and insoluble particles and
can exist in the Aitken, accumulation and coarse modes. BC aerosols are assumed to be 100%
hydrophobic and in the Aitken mode upon emission. Aging then occurs due to condensation of
sulfuric acid produced in the gas-phase reaction $OH+SO_2 \longrightarrow H_2SO_4$ or coagulation with sulfate
particles. M7 is coupled to the sulfur/oxidant chemistry in the OsloCTM2, i.e., the production
of sulfate is explicitly calculated and is dependent on the $SO_2$ emissions and oxidant levels and
thus variable in time and space.



Particles in the soluble modes are assumed to be hygroscopic and are removed according to the
fraction of the liquid plus ice water content of a cloud that is removed by precipitation (Berntsen
et al., 2006), assuming 100% scavenging efficiency by both water and ice in both large-scale
and convective precipitation in the baseline setup. Since Lund and Berntsen (2012) the temporal
frequency of wet scavenging in OsloCTM2-M7 has been reduced from three to one hour. The
dry deposition velocities for all aerosols depends on particle size and density, turbulence close
to surface and the resistance of the laminar sub layer (Grini, 2007). The OsloCTM2-M7 also
keeps track of the BC concentration in snow. Snow depth and snowfall data from ECMWF is
used to build snow layers in the model and BC is dry and wet deposited in these. For detailed
description see Appendix A of Skeie et al. (2011).
The sulfate and nitrate modules are described in detail in Berglen et al. (2004) and Myhre et al.
(2006), and we only give brief summaries here.
The sulfur cycle chemistry scheme includes dimethyl sulfide (DMS), $SO_2$, sulfate, $H_2S$ and
methane sulfonic acid (MSA) and the concentrations of sulfur is calculated interactively with
the oxidant chemistry. Gas-phase oxidation of $SO_2$ by OH forms sulfate and $SO_2$ is also
oxidized to aqueous phase sulfate by $H_2O_2$, $HO_2NO_2$ and $O_3$. When M7 is used, the gas-phase
sulfate is saved in a separate tracer and allowed to condense on the insoluble aerosol modes.
The aqueous phase sulfate is transferred to the accumulation and coarse mode sulfate tracers in
M7 according to a prescribed fraction. The treatment of sulfate aerosols then follows M7.
The chemical equilibrium of semi-volatile inorganic species is treated with the Equilibrium
Simplified Aerosol model (EQSAM) (Metzger et al., 2002a; Metzger et al., 2002b). EQSAM
considers the $NH_4^+$/$Na^+$/$SO_4^{2-}$/$NO_3^-$/$Cl^-$/$H_2O$ system and calculates the gas/aerosol partitioning
of ammonium nitrate under the assumption that aerosols are internally mixed and obey
thermodynamic gas/aerosol equilibrium. Nitrate aerosol is represented by two modes; a fine
mode comprised of sulfate and a coarse mode comprised of sea salt. After $H_2SO_4$ and $HNO_3$
have been generated by the photochemistry, the thermodynamic equilibrium is solved using
EQSAM.

### 2.2 Emissions

Anthropogenic emissions for 2008-2010 are from the ECLIPSEv4 inventory developed with
the GAINS model (Amman et al. 2011) as part of the activities under the ECLIPSE project
funded by the European Commission 7[th] Framework (Amann et al., 2011; Klimont et al., 2009;





Klimont et al., 2016) (available upon request from http://eclipse.nilu.no/). Emissions from
international shipping and aviation are from the Representative Concentration Pathway (RCP)
6.0 (Fujino et al., 2006; Hijioka et al., 2008). Biomass burning emissions are from the Global
Fire Emission Database version 3 (GFEDv3) (van der Werf et al., 2010). Seasonal variability
in domestic emissions is accounted for by using monthly weights (2000-2006 average) for each
grid based on spatially distributed temperature data from the Climate Research Unit (CRU)
following the methodology described in Streets et al. (2003). Total BC emissions in 2010 are
5866 Gg from fossil fuel plus biofuel sources and 2273 Gg from biomass burning.
*2.3 Experiments*
We first perform a three-year base simulation with meteorological data and emissions for 2008-
2010, which forms the basis for the model evaluation. Next, we perform a range of sensitivity
experiments described in the following paragraphs and summarized in Table 1.

Several sensitivity experiments are related to the aging of BC. First, we explore the impact of
varying the amount of soluble material required to transfer the BC aerosols to the soluble mode.
The M7 uses the concept of monolayers (ML); when sufficient soluble material is associated
with a hydrophobic particle to form "n" monomolecular layers around the particle, the particle
is assumed to be hygroscopic and is transferred to the mixed mode. Currently, n=1 is used based
on the best agreement with a sectional model found by Vignati et al. (2004). However, the
amount of soluble material required for a particle to become hygroscopic is an important source
of uncertainty (Vignati et al., 2010). Popovicheva et al. (2011) used a laboratory approach to
quantify the water uptake by particles with varying amounts of sulfates in order to simulate the
aging of combustion particles. Based on a quantification measure for separating hygroscopic
and non-hygroscopic soots (Popovicheva et al., 2008), the laboratory results suggest that the
transformation of soot particles from hydrophobic to hydrophilic requires an $H_2SO_4$ surface
coverage of 0.5-1.4 ML, while 1.4-2.3 ML were required for transformation to hygroscopic
mode. Based on these results we perform three model simulations assuming 0.5, 1.4 and 2.3
ML required for BC aging. Next, we perform a test where 50% of BC from biomass burning
sources is emitted directly in the accumulation mode instead of in the insoluble Aitken mode.
This is based on observational evidence suggesting that biomass burning BC tends to be larger
and more aged, with thicker coatings than BC from urban source (Schwarz et al., 2008). Finally,
we test the impact of allowing for BC aging by condensation of nitric acid ($HNO_3$), first
including total $HNO_3$ and then excluding $HNO_3$ produced in the aqueous-phase reaction with



$N_2O_5$. We extend, in a simplified manner, M7 to also account for condensation by $HNO_3$ on
insoluble particles after gas/aerosol partitioning with ammonium-nitrate is calculated in
EQSAM. We follow the same treatment of condensation as for sulfate in M7 (Vignati et al.,
2004) and adopt an accommodation coefficient for $HNO_3$ of 0.1 (Pringle et al., 2010). Three
different runs are performed where the number of required ML are assumed to be one (as for
sulfate in the standard M7 case), five or ten. Results presented in Sect. 3 uses ML=5.

The second set of sensitivity tests is related to emissions and wet scavenging, the main loss
mechanism of BC and hence a key parameter for the lifetime and distribution. Hydrophilic BC
is originally assumed to be 100% removed by both liquid and ice in large-scale mixed-phase
clouds in the OsloCTM2-M7. However, this high efficiency of BC removal by ice-phase
precipitation is uncertain. Koch et al. (2009a) found that assuming 12% ice removal of BC gave
optimal agreement with observations. This fraction was also supported by observations in Cozic
et al. (2007) and has been adopted in studies with the OsloCTM2 bulk aerosol parameterization
(e.g. Skeie et al. (2011)). Here we compare results with 100% and 12% removal efficiency for
large-scale ice-phase clouds. The removal scheme in OsloCTM2-M7 also assumes no wet
scavenging of hydrophobic particles. However, hydrophobic BC aerosol may still be subject to
removal by impact scavenging or act as ice nuclei (IN) in convective and mixed-phase clouds
(Ekman et al., 2006; Kajino et al., 2012; Park et al., 2005). The BC IN activity is not well known.
In order to represent at least some of this uncertainty, we perform two sensitivity tests assuming
either 100% or 20% removal efficiency of hydrophobic BC by convective precipitation, with
the latter loosely based onHoose et al. (2010). We also perform a combination test assuming
12% removal efficiency of hydrophilic BC by large-scale ice-phase clouds and 20% removal
of hydrophobic BC by convective precipitation.

Finally, we perform two additional tests to investigate the impact of seasonality in domestic
and agricultural waste burning emissions and higher emissions in Russia following a recent
study by Huang et al. (2015). In the first, we alternately remove the seasonal variation in
domestic and agricultural waste burning emissions, while in the second the ECLIPSEv4
emissions in Russia are replaced by Huang et al. (2015). These have limited impact on the
global BC distributions, but their influence on the seasonal cycle of Arctic BC concentrations
is discussed in Sect. 3.1.1.

TABLE 1

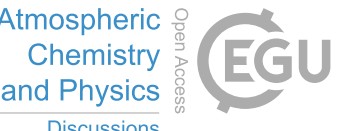

### 2.4 Radiative forcing and temperature response

To estimate implications of the concentration changes in our experiments for the global BC climate impact, we use 3-dimensional, temporally varying radiative forcing (RF) and surface temperature response (TS) kernels derived from simulations with the CESM-CAM4, where a uniform BC burden was systematically added to each model layer to investigate the climate response to a BC perturbation at a given altitude (Samset & Myhre, 2015). Because the BC perturbations were applied uniformly throughout a single model layer, the temperature response at each grid point is caused partly by the BC forcing exerted locally and partly by forcing in surrounding gridboxes. For each experiment, we therefore multiply the globally averaged vertical BC profile from the OsloCTM2-M7 with the globally averaged forcing and temperature change kernels, respectively. Both direct and semi-direct effects due to aerosol-radiation interactions are included in the kernel response. In line with the nomenclature of the IPCC Fifth Assessment Report we refer to the net effect as effective radiative forcing (ERFari) and the direct effect only as RFari.

CAM4 does not account for the absorption enhancement due to BC aging, resulting in a lower direct RF per BC burden than earlier studies, especially at higher altitudes (e.g., Samset and Myhre (2011)). The consequent temperature response per unit BC may also be underestimated. However, here we focus on the changes from the baseline in each sensitivity experiment rather than absolute climate impacts.

### 2.5 Observations

Modeled concentrations are evaluated against measurements from surface stations, flight campaigns and snow pack samples.

Measured surface concentrations of BC, sulfate, nitrate, sulfur dioxide and nitric acid across North America are from the IMPROVE and CASTNET networks, while measurements across Europe and the rest of the world are from the EBAS and NOAA GMD databases. We also compare against measurements in China from Zhang et al. (2012) and from Aerosol Mass Spectrometer (AMS) campaigns summarized in Zhang et al. (2007).

To evaluate the model performance we calculate the correlation coefficient and the mean normalized bias (MNB). The MNB for each species is given by Equation 1:





$$MNB = \frac{1}{N} \sum \left( \frac{C_{mod} - C_{obs}}{C_{obs}} \right) \qquad (1)$$

where $C_{mod}$ and $C_{obs}$ is modeled and observed concentration and N is the total number of
observations.
Following the recommendations by Petzold et al. (2013) observational data are referred to as
equivalent BC (EBC), refractory BC (rBC) or elemental carbon (EC) depending on whether
measurements are derived from optical absorption methods, incandescence methods or methods
that specify the carbon content in carbonaceous matter. To convert to BC concentrations we
adopt a mass-absorption cross-section (MAC) of 9.7 m$^2$/g (Bond & Bergstrom, 2006) for all
stations except Alert and Zeppelin, where we use the MAC given in Lee et al. (2013).
BC in snow is compared to snow sample measurements across the Arctic in 2008/2009 (Doherty
et al., 2010) and across Northern China in 2010 and 2012 (Wang et al., 2013; Ye et al., 2012).
In the latter case, model results for 2010 are used.
Vertical profiles of modeled BC is compared with measurements from several flight campaigns,
including ARCPAC (Aerosol, Radiation, and Cloud Processes affecting Arctic Climate),
ARCTAS (Arctic Research of the Composition of the Troposphere from Aircraft and Satellites),
HIAPER Pole-to-Pole Observations (HIPPO) and A-FORCE (Aerosol Radiative Forcing in
East Asia). During ARCPAC and ARCTAS, flights were made across Alaska and Canadian
Arctic in spring and summer of 2008 (Brock et al., 2011; Jacob et al., 2010), while HIPPO
measured atmospheric constituents along transects from approximately pole-to-pole over the
Pacific Ocean during different seasons from 2009 to 2011 (Wofsy et al., 2011). The A-FORCE
campaign sampled air masses around Japan in March-April 2009 (Oshima et al., 2012). Data
from    ARCPAC,    ARCTAS    and    HIPPO    is    available    online    from
www.esrl.noaa.gov/csd/projects/arcpac/,    www.air.larc.nasa.gov/missions/arctas/arctas.html
and hippo.ornl.gov/. Data from A-FORCE was provided by Professor Yutaka Kondo,
University of Tokyo (personal communication). Model data is also compared with CO
concentrations measured during the campaigns.
Model data is interpolated in time and space and extracted along the flight track. An average
profile for each campaign and latitude band is calculated by averaging observations and model
results in 100 hPa altitude bins (25 hPa for HIPPO data between 400 and 200 hPa). The HIPPO
data is also separated into five latitude bands. To evaluate the model performance in each



experiment, we calculate the MNB for each campaign following Eq. 1, where N is determined
by the number of altitude and latitude bins.

## 3 Results and discussion

**3 Results and discussion**
*3.1 Model evaluation*
We first evaluate the general performance of the OsloCTM2-M7. While the main focus of this
paper is BC, the evaluation is extended to include species relevant for the BC aging process,
including sulfate and sulfur dioxide. We also look at the modeled CO distribution. CO is another
product of incomplete combustion and therefore has many of the same emission sources as BC.
However, due to the longer lifetime of CO a comparison with observations, in particular in the
more remote regions mainly influenced by long-range transport, can give an indication of how
well the model represents the atmospheric transport.

### 3.1.1 Surface concentrations

*3.1.1 Surface concentrations*
Figure 1 shows annual mean (year 2008) modeled and measured surface concentrations of BC,
sulfate, nitrate, sulfur dioxide and nitric acid.
FIGURE 1
The OsloCTM2-M7 underestimates BC and sulfate surface concentrations, with MNB of -0.55
and -0.45, respectively. The underestimation is largest for measurements in China. Nitrate
concentrations are in better agreement with measurements, with MNB of 0.08. The model
overestimates surface concentrations of sulfur dioxide, especially in Europe, with MNB of 0.70.
This may be due to too inefficient conversion to sulfate, which is supported by the
underestimation of sulfate aerosols, and/or an overestimation of emissions. Also nitric acid
concentrations in Europe and North America are overestimated (MNB 0.75).
We also investigate the seasonal cycle of BC. Figure 2 shows monthly mean modeled BC and
measured EBC surface concentrations averaged over 2008-2010. The model captures the
magnitude relatively well at Mace Head, Cape Point, Trinidad Head, Barrow and Pallas, but
fail to capture some of features of the seasonal variation. Concentrations are also
underestimated at Lulin, Hohenpeissenberg and Jungfraujoch during winter and spring.
FIGURE 2



Many models typically struggle to capture the seasonal cycle and magnitude of measured high-
latitude BC surface concentrations. While there has been considerable progress and current
models capture high-latitude seasonality better than previous generations (Breider et al., 2014;
Browse et al., 2012; Liu et al., 2011; Sharma et al., 2013), problems remain. This is also the
case for the OsloCTM2-M7. Lund and Berntsen (2012) showed that inclusion of aerosol
microphysics significantly improved both magnitude and seasonality of Arctic BC. This is
further improved by the use of updated emissions in the current study, partly due to the inclusion
of emissions from flaring, which is an important local Arctic source of BC (Stohl et al., 2013).
However, the model still underestimates concentrations during spring. The seasonal variability
in emissions is an important factor. Accounting for seasonality in domestic BC emissions in the
ECLIPSEv4 inventory increases the burden of total fossil fuel plus biofuel BC north of 65°N
by approximately 15% during winter and by 2% on annual average compared to assuming
constant monthly emissions. Over the same region, including seasonality in agricultural waste
burning results in a 2-3% higher total BC burden during spring. This is a relatively small
increase, but agricultural waste burning contributes only around 6% to total BC emissions north
of 40°N on an annual basis. Another potentially important factor is missing or underestimated
emission sources. A recent study by Huang et al. (2015) estimate total anthropogenic BC
emissions in Russia of 224 Gg, about 40% higher than in the ECLIPSEv4 inventory. Replacing
the Russian BC emissions in the ECLIPSEv4 inventory with those from Huang et al. (2015)
increases the modeled BC burden north of 65N by about 16% during fall, winter and early
spring and 2-10% during summer. Another possibly underestimated emission source is open
waste burning. Wiedinmyer et al. (2014) estimate that 631 Gg BC is emitted globally from open
waste burning, nearly a factor 7 more than in the ECLIPSEv4 inventory. Moreover, they suggest
that open waste burning may contribute 30-50% to total anthropogenic $PM_{10}$ emissions in
Russia, from where the near-surface transport of BC to the Arctic is effective (Stohl, 2006).
Underestimation of this emission source may thus contribute to the too low modeled Arctic BC
concentrations.
Eckhardt et al. (2015) show that models, including the OsloCTM2, have similar difficulties
capturing the sulfate seasonality in the Arctic as they have for BC. At Zeppelin, the OsloCTM2-
M7 underestimates also sulfur dioxide during spring, but overestimates concentrations during
summer.
Figure 3 shows the seasonal cycle of CO for the same stations as in Fig. 2. In the Northern
Hemisphere, the model captures the measured concentrations during summer, but



underestimates the magnitude during winter/spring, a feature that has been shown also for other
models in previous studies (Emmons et al., 2015; Monks et al., 2015). We also compare results
at additional Southern Hemisphere locations (not shown here). In the Southern Hemisphere, the
model generally reproduces the magnitude better, with a slight overestimation during
winter/spring at several stations. The ability of the model to reproduce the seasonal cycle and
magnitude of CO, in particular at remote Southern Hemispheric stations that are mainly
influenced by long-range transport, suggests that the model represents atmospheric transport
reasonably well and points to other processes as the dominant source of uncertainty in the model.
FIGURE 3
*3.1.2 Vertical profiles*
Figures 4 and 5 show modeled vertical BC and CO profiles against measurements from six
aircraft campaigns. Compared to measurements from ARCPAC and ARCTAS spring the
OsloCTM2-M7 underestimates the magnitude of BC concentrations throughout the atmosphere
(Fig. 4 (p),(r); MNB -0.8). During both these campaigns, air masses were heavily influence by
biomass burning plumes, which are often not captured by global models. The same springtime
discrepancy was also seen in the surface concentrations. However, the shape of the profile is
reproduced reasonably well. The agreement is better for ARCTAS summer (Fig. 4 (q); MNB
0.05), but the model underestimates near-surface concentrations. The model also
underestimates the magnitude of CO concentrations during these two campaigns (Fig. 5 (p-r)),
but again captures the profile shape reasonably well, providing further indication of too low
emissions as an important source of the discrepancy.
Measurements from HIPPO are separated into five latitude bands (Fig. 4 (a-o), Fig. 5 (a-o)).
For most latitude bands and flights, there is reasonable agreement close to the surface. In the
60-80N latitude band, the model overestimates concentrations close to the surface during
HIPPO1 and 2, but underestimates concentrations during HIPPO3. HIPPO3 was undertaken
during spring and a similar underestimation was also seen in the modeled surface measurements
at Barrow during this time of year (Fig. 2). The model typically fail to reproduce the layered
structure of the measured vertical profiles. In particular the high-altitude concentrations in
tropics and the southern mid-latitudes are overestimated. It should be noted that there are
substantial differences between the three HIPPO campaigns although they all cover the Pacific.
A better model-measurement agreement is found for HIPPO3 than for HIPPO1 and 2 (MNB



1.1, 3.3 and 2.8, respectively). In contrast to BC, both the magnitude and shape of most vertical
CO profiles compare well across all latitude bands
There is quite good agreement between measured and modeled BC and CO during the A-
FORCE campaign (Fig. 4 (s), Fig. 5 (s); MNB -0.1), with model results falling within one
standard deviation of the measured profile. The A-FORCE campaign was carried out
downstream of nearby emission sources and the good agreement with observations suggests
reasonable representation in the model of both emission magnitude in the region and the mixing
with the free troposphere on timescales of a few days. On these temporal and spatial scales, the
loss processes are of less importance for the aerosol distribution. In contrast, the HIPPO
campaigns sampled older air masses and loss processes are more important.
Our overall findings are in line with other recent studies. The tendency to overestimate high
altitude BC concentrations over the Pacific has been noted for several other model (Kipling et
al., 2013; Samset et al., 2014; Schwarz et al., 2013; Wang et al., 2014). The vertical profiles
from OsloCTM2-M7 also fall roughly within the range of the AeroCom Phase II models
(Samset et al., 2014).
FIGURE 4
FIGURE 5

### 3.1.3 BC in snow

The OsloCTM2-M7 underestimates BC concentrations in snow compared to measurements, in
particular in Russia, Svalbard and the Canadian Arctic. Here we find somewhat higher modeled
concentrations than in previous studies (Lund & Berntsen, 2012; Skeie et al., 2011) owing to
the updated emission inventory and shorter model time step for precipitation. However, this
increase is not sufficient to fully compensate for the existing underestimation. The model and
measurements agree better for many of the snow samples taken in China.
Towards late spring, the modeled concentrations are occasionally very high compared to the
measurement, especially in Tromsø and the Arctic Ocean. This feature was also shown by Skeie
et al. (2011). During melting, the model assumes that all BC accumulates at the surface.
Observational evidence suggest this assumption may lead to an overestimation. For instance,
scavenging fractions of 10-30% due to percolation of meltwater were found by Doherty et al.
(2013) from measurements made in Alaska, Greenland and Norway during melt season.




### 3.2 Sensitivity of BC concentrations to changes in aging and scavenging

This section discusses the sensitivity of modeled BC concentrations to the changes in aging and
scavenging processes in our experiments.
Table 2 summarizes the global BC burden and lifetime in each experiment. The global mean
burden (lifetime) is 133 Gg (6 days) in the base simulation, while there is considerable range
from 81 Gg (3.6 days) to about 185 Gg (8 days) across the experiments. This range still falls
within that of BC lifetimes from global models (e.g., Samset et al. (2014)).
TABLE 2
The largest changes in BC concentrations in the sensitivity experiments occur in remote regions
and we find only small differences in the model-measurement comparison at the more
urban/rural stations in Fig. 2. In the following we therefore focus on the Arctic stations (Alert,
Barrow, Pallas and Zeppelin), as well as the vertical profiles from the six aircraft campaigns.
Figure 6 shows seasonal Arctic surface concentrations compared to the measurements (left
column) and the absolute difference from the base in each experiment (right column). Figure 7
shows the vertical BC profiles for each campaign and experiment, compared to the baseline and
measurements.
FIGURE 6
FIGURE 7
A shorter atmospheric BC lifetime reduces the high-altitude overestimation at mid- and tropical
latitudes over the Pacific. This is in line with other recent studies, which have suggested that
the lifetime of BC in global models must be reduced in order for the models to reproduce the
HIPPO data (Hodnebrog et al., 2014; Samset et al., 2014; Wang et al., 2014). Both allowing for
convective scavenging of hydrophobic BC (ConvBCi) and reducing the amount of soluble
material required for aging (CoatThick0.5) substantially reduces the MNB for the HIPPO
campaigns compared to the baseline (from approx. 3 to -0.3 and 1, respectively). For vertical
profiles in most latitude bands, the former experiment results in the lowest MNB of the two.
However, the model is very sensitive to the fraction of hydrophobic BC assumed to be available
for removal (here 100% or 20%), which is an uncertain parameter. Surface concentrations at
Alert, Zeppelin, and Pallas are also reproduced reasonably well in these experiments, although




the springtime underestimation discussed above remains. In other parts of the Arctic however,
the model performance is exacerbated. More specifically, the MNB for the ARCTAS and
ARCPAC campaigns increases and the underestimation of surface concentration at Barrow is
larger compared to the baseline. Similar effects are also found in the 60°-70°S region (Fig. 7
(e), (j)). In addition to aging and scavenging, several other factors likely contribute to the too
low modeled Arctic concentrations, including uncertainties in emissions and model resolution.
A recently published study point to the importance of model resolution as a source of
uncertainty, suggesting that a kilometer-order resolution is required for more accurate
representation of BC concentrations in the Arctic (Sato et al., 2016).

Conversely, increasing the amount of soluble material required for aging increases the BC
lifetime. This in turn results in an increased potential for long-range transport and increase in
Arctic surface concentrations. However, with the exception of Barrow during spring, increasing
the number of required ML (CoatThick1.4, CoatThick2.3) does not result in marked
improvements in modeled Arctic surface concentrations compared to measurements. The
longer aging time in these experiments also results in a poorer agreement with the HIPPO
measurements, both close to the surface and at high altitudes. Moreover, even with the longer
lifetime and consequent increases in Arctic BC concentrations, the model does not reproduce
the vertical profiles from ARCTAS and ARCPAC. The experiments also result in reduced
concentrations of BC in snow in our model. In these cases, the aging time is longer and more
BC hence resides in the insoluble mode, unavailable for wet scavenging. Hence, in the
OsloCTM2-M7 a slower BC aging alone does not result in significant improvements in model-
measurement discrepancies.

Reducing the scavenging of BC by large-scale ice clouds and increasing the fraction of biomass
burning emissions initially in the accumulation mode, have only a minor influence on both
Arctic surface concentrations and modeled vertical profiles compared to the baseline. This is
also the case for the combined reduction in scavenging by large-scale ice clouds and increased
convective scavenging of hydrophobic aerosols.

In terms of BC concentrations in snow, smaller improvements are found, but none of the
experiments improve the model-measurement comparison of BC in snow simultaneously in all
regions.





Measurements at mid-latitudes have shown that nitrate is frequently present in internal aerosol
mixtures and contribute to the aging of BC (Pratt & Prather, 2010; Shiraiwa et al., 2007). The
addition of nitric acid in the microphysical BC parameterization is a novel treatment in the
CTM2-M7 and these experiments are discussed separately here. Allowing for nitric acid to
condense on the aerosols results in a faster aging as more soluble material is available than
when only sulfate is allowed to contribute and hence reduces the global BC lifetime. This in
turn reduces high-altitude BC concentrations and leads to a better agreement with the HIPPO
measurements (MNB between 0.4 and 0.7 for HIPPO1 and 2 in NitCondv2). Furthermore, BC
snow concentrations across all regions except Greenland increase in this experiment, although
not enough to fully account for the existing underestimation compared to measurements.
However, the Arctic atmospheric BC concentrations are significantly reduced, resulting in a
poorer model performance compared to both measured vertical profiles and surface
concentrations in this region.

In this study, we have taken a first step towards inclusion of nitrate in the microphysical aerosol
parameterization. This should however be studied further in future work. For instance the
current setup only treats the condensation by nitric acid, not coagulation with nitrate aerosols.
Furthermore, it is uncertain how effectively nitric acid increases the hygroscopicity of BC. Here
we have assumed 5 ML. In two additional sensitivity tests we also investigate the impact of 1
and 10 ML and find substantial impacts on modeled BC concentrations. Existing model-
measurement discrepancies in nitrate and sulfate concentrations also contribute to uncertainties.

In this work, we have not considered combinations of or regionally differing sensitivity
experiments, for instance increased coating thickness required at high-latitudes in combination
with more efficient removal by convective precipitation in low and mid-latitudes. Moreover,
there are important details that are not captured in the OsloCTM2-M7. One example is related
to the particle hydrophilicity/hygropscoicity. The OsloCTM2-M7 assumes that particles can
automatically act as cloud condensation nuclei once they are transferred from the hydrophobic
to hydrophilic mode. However, the cloud condensating activity of hydrophilic and hygroscopic
particles also depends on the atmospheric supersaturation (Koehler et al., 2009; Petters &
Kreidenweis, 2007). Furthermore, particles may not merely be hydrophilic or not as assumed
by models, but rather exhibit a whole range of degrees of hydrophilicity. The ice nucleating
efficiency of BC is also relatively poorly known. Our results underline the importance of more





observations, in particular of the mixing state and scavenging of BC, as well as experimental
data to improve process understanding.

### 3.3 Climate impacts

The changes in BC concentrations in our experiments can in turn affect the climate impact,
especially when changes occur at altitudes where the efficacy of BC forcing and temperature
response is strong. Changes in global radiative forcing (RF) and surface temperature (TS) from
the baseline in each experiment are estimated using a kernel approach based on results from
Samset and Myhre (2015) (see Sect. 2.4) and presented in Table 2.
Relative to the baseline, a decrease in global-mean BC ERFari (i.e., net of direct and semi-
direct aerosol-radiation interactions) of -49 and -45 mW/m$^2$ is estimated for the two
experiments that lead to the most marked improvements (i.e., strongest reduction in MNB) in
vertical profiles compared to measurements over the Pacific (ConvBCi and NitCondv2). In
these two experiments, allowing for convective removal of hydrophobic BC and adding
condensation by gas-phase nitric acid reduces BC concentrations at high altitudes where the
forcing efficacy is strong. Reducing the amount of sulfate required for BC aging also gives a
notable decrease in ERFari of -26 mW/m$^2$. Changes in ERFari of similar magnitudes but
opposite sign are estimated for the CoatThick1.4 and CoatThick2.3 experiments. The change
in surface temperature response is also largest for three former experiments, resulting in a
decrease of -25 mK compared to the baseline.
To place the impact of our experiments in context, we calculate the change in direct forcing
only (i.e., RFari) and compare with existing best estimates of the total pre-industrial to present
BC RFari. The Fifth IPCC Assessment Report reports a best estimate of RFari due to BC from
all sources of 0.6 W/m$^2$ (Boucher et al., 2013), while Bond et al. (2013) give a slightly higher
estimate of 0.71 W/m$^2$. Depending on experiment, the changes estimated here are on the order
of 10 to 30% of the total BC RFari relative to pre-industrial.
Since our study also focuses on Arctic BC, we estimate the change in ERFari and TS caused by
the changes in the BC profiles over this region. The resulting ERFari changes are generally
larger than in the global-mean case. For all except two cases the Arctic TS changes are also
larger than the global-mean changes. This partly reflects the large BC concentration changes in
this region in our experiments and partly a smaller contribution of the semi-direct effect to the
ERFari, which acts to offset less of the RFari than on global average. The Arctic surface





temperature response to BC forcing exerted in the lower atmosphere, where a substantial impact
on BC concentrations is seen in several of the experiments, is also stronger than in lower
latitudes.
There is, however, an important caveat when using the temperature kernel from Samset and
Myhre (2015) to estimate Arctic impacts. Because the BC perturbations at each altitude were
applied uniformly in that model layer, the impact on temperature in a specific gridbox may be
due both to forcing exerted locally and to remote forcing through large-scale circulation impacts.
To exclude any influence of BC forcing exerted outside the Arctic region, we also use results
from Flanner (2013) to estimate the TS changes. Using the same model as Samset and Myhre
(2015), Flanner (2013) imposed BC perturbations at five different altitudes over the Arctic only,
hence calculating the Arctic TS to only local effects. The resulting temperature kernel has
previously been used to assess the impact of regional on-road diesel BC emissions (Lund et al.,
2014). When used here to estimate the impact of our experiments, we find similar changes in
Arctic TS to those estimated using results from Samset and Myhre (2015), with one notable
exception. In three of the experiments (EmisTest, LSice12 and CombPert) the two different
approaches produce changes in net Arctic TS of opposite sign. This is caused by slightly
different efficacies in the two temperature kernels above 500 hPa altitude, where these
experiments have their largest effect on BC concentrations. For most of the remaining
experiments, using the temperature kernel from Samset and Myhre (2015) result in slightly
stronger changes in net Arctic TS, reflecting the higher efficacy below 850 hPa compared to
the Flanner (2013) kernel.
**4 Summary and conclusions**
We have performed a range of experiments to investigate the sensitivity of BC concentrations
modeled by the OsloCTM2-M7 to parameters controlling the aerosol scavenging and aging and
how these processes influence the existing model-measurement discrepancies, focusing
simultaneously on Arctic surface concentrations and remote vertical distributions of BC. The
experiments include a novel treatment of condensation of nitric acid on BC. Furthermore, the
subsequent impact of concentration changes on radiative forcing and surface temperature
response is estimated.

The OsloCTM2-M7 underestimates annual averaged BC surface concentrations, with a mean
normalized bias (MNB) of -0.55. The model is better able to reproduce the observed seasonal





variation and magnitude of Arctic BC surface concentrations compared to previous OsloCTM2 studies, but model-measurement discrepancies remain, particularly during spring. The OsloCTM2-M7 overestimates high-altitude BC concentrations over the Pacific compared to measurements from the HIPPO flight campaign, as has been found also for several other global models.

We find that a shorter atmospheric BC lifetime in the model reduces the high-altitude overestimation at mid- and tropical latitudes over the Pacific. This is in line with other recent studies which have suggested that the lifetime of BC in global models must be reduced in order for the models to reproduce the HIPPO data (Hodnebrog et al., 2014; Samset et al., 2014; Wang et al., 2014). Both allowing for convective scavenging of hydrophobic BC and reducing the amont of soluble material required for aging significantly improves (i.e., reduces the MNB) the comparison with vertical profiles from HIPPO compared to the baseline. In the case of convective scavenging, the model is sensitive to the fraction of hydrophobic BC assumed to be available for removal, a parameter with large associated uncertaines. While the surface concentrations at the Arctic stations of Alert, Zeppelin and Pallas remain in reasonable agreement with observations in the two former experiments, the comparison with measurements at Barrow and the ARCTAS and ARCPAC fligh campaigns becomes poorer. Conversely, changes in processes that lead to a longer BC lifetime excacerbates the high-altitude overestimation over the Pacific and result in an overestimation of Arctic surface concentrations during winter. Moreover, despite increases compared to the baseline, the BC concentration in snow and during flight campaigns in the Arctic is still underestimated.

Measurements at mid-latitudes have shown that nitrate is frequently present in internal aerosol mixtures and contribute to the aging of BC (Pratt & Prather, 2010; Shiraiwa et al., 2007). In this study, we have taken a first step towards including this process in the OsloCTM2-M7 by allowing for aging of BC by condensation of nitric acid. This results in a faster aging and hence a reduced global lifetime, which in turn reduces high-altitude BC concentrations and leads to a better agreement with the HIPPO measurements. Furthermore, BC snow concentrations across all regions except Greenland increase in this experiment, although not enough to eliminate the underestimation compared to measurements. However, the Arctic atmospheric BC concentrations are substantially reduced, resulting in a poorer model performance compared to both measured vertical profiles and surface concentrations in this region. A number of





uncertainties remain, including how effectively nitric acid increases the hygroscopicity of BC
and how coagulation with nitrate aerosols influence aging, and should be studies further.

Our experiments result in a non-negligible impact on radiative forcing (RF) and surface
temperature (TS). Compared to the baseline, decreases in the global RFari (i.e., direct RF) on
the order of 10-30% of the total pre-industrial to present BC direct forcing is estimated for the
experiments that result in the largest changes in BC concentrations. Notable decreases in both
ERFari (i.e., direct plus semi-direct RF) and TS is also estimated for the experiments which
leads to the most marked improvements (i.e., strongest reduction in MNB) in vertical BC
profiles compared to measurements over the Pacific.

While we find that globally tuning parameters related to aging and scavenging can improve the
representation of BC in the OsloCTM2-M7 compared to measurements in specific regions, our
results also show that such improvements can result from changes in several processes and
dependen on assumptions about uncertain parameters such as the ice nucleating efficacy of BC
and the change in hygroscopicity with aging. It is also important to be aware of potential
tradeoffs in model performance between different regions. Other important sources of
uncertainty, particularly for Arctic BC, such as model resolution has not been investigated here.
Our results underline the importance of more observations and experimental data to improve
process understanding and thus further constrain models.

**Acknowledgements**
This work was funded by the Research Council of Norway through the projects TEMPO,
SLAC and AC/BC. We also acknowledge the Reseach Council of Norway's programme for
supercomputing (NOTUR). We thank Professor Yutaka Kondo, University of Tokyo, for
providing results from the A-FORCE flight campaign.







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

**TABLES**
*Table 1: Summary and description OsloCTM2-M7 experiments performed in this study.*

| Experiment | Description |
|---|---|
| Baseline | Standard M7 OsloCTM2 simulation |
| CoatThick0.5 | Required coating thickness reduced to 0.5ML |
| CoatThick1.4 | Required coating thickness increased to 1.4ML |
| CoatThick2.3 | Required coating thickness increased to 2.3ML |
| EmisTest | 50% of biomass burning BC emitted directly in soluble accumulation mode |
| ConvBCi100 | Hydrophobic BC removed by convective precipitation, 100% efficiency |
| ConvBCi20 | Hydrophobic BC removed by convective precipitation, 20% efficiency |
| LSice12 | Scavenging by ice in large-scale precipitation reduced from 100% to 20% |
| CombPert | LCice12 + ConvBCi20 |
| NitCond | Aging by HNO3 condensation included |
| NitConcV2 | As above, but excluding HNO3 produced by aqueous-phase $N_2O_5$ reaction |
| EmisSeasonality | Seasonality in domestic or agricultural waste burning BC emissions removed |
| EmisBCRUS | BC emissions in Russia replaced by Huang et al. (2015) inventory |



*Table 2: BC lifetime and burden, and the change in global-mean RF, DRF and surface*
*temperature response from the baseline in each experiment.*

| | Global | | | | |
|---|---|---|---|---|---|
| | Lifetime | Burden | ΔERFari | ΔRFari | ΔTS |
| | [days] | [Gg] | [mW/m²] | [mW/m²] | [mK] |
| **Base** | 6.0 | 133 | - | - | - |
| **CoatThick0.5** | 4.8 | 106 | -26 | -88 | -14 |
| **CoatThick1.4** | 6.7 | 150 | 18 | 55 | 11 |
| **CoatThick2.3** | 8.3 | 185 | 52 | 166 | 32 |
| **EmisTest** | 5.9 | 131 | -1 | -7 | -0.3 |
| **ConvBCi100** | 3.6 | 81 | -49 | -181 | -25 |
| **ConvBCi20** | 4.8 | 107 | -24 | -91 | -11 |
| **LSice12** | 6.6 | 147 | 15 | 46 | 8 |
| **Combpert** | 6.6 | 148 | 15 | 49 | 8 |
| **NitCond** | 4.9 | 109 | -24 | -84 | -11 |
| **NitCondv2** | 3.9 | 87 | -45 | -157 | -23 |










**FIGURES**

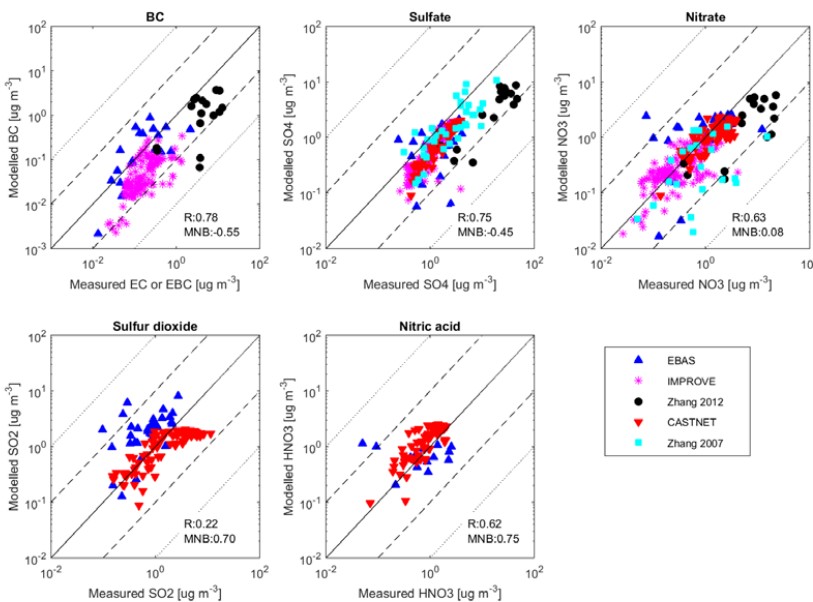


*Figure 1: Annual mean measured versus modelled BC, sulfate, nitrate, sulfur dioxide and nitric*
*acid surface concentrations across Europe, North America and Asia.*





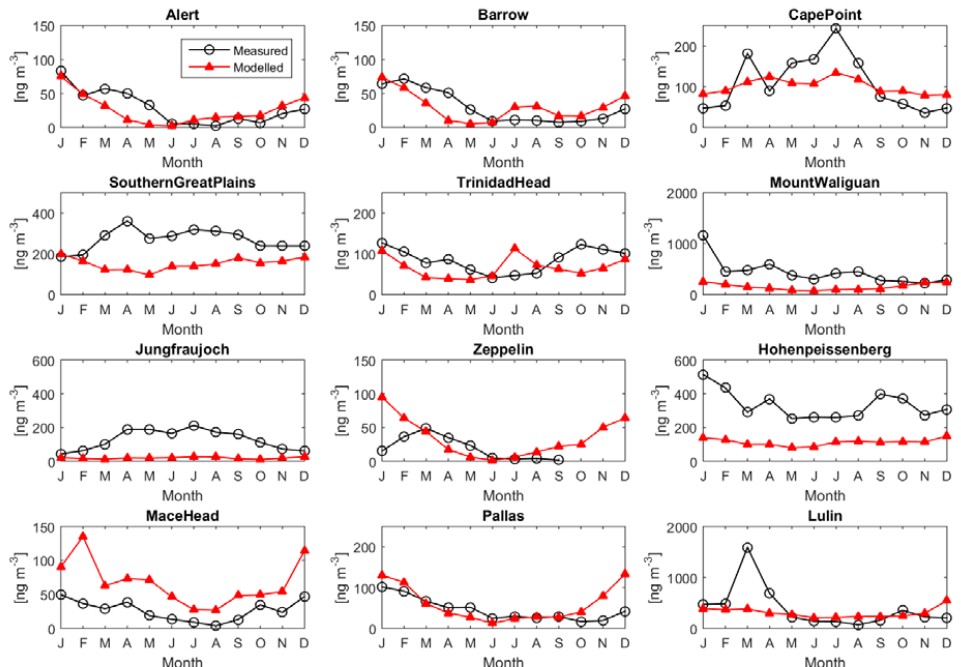


*Figure 2: Monthly mean measured EBC versus modelled BC surface concentrations [ng/m3]*
*averaged over 2008-2010 (data at Lulin only available for 2009-2010).*















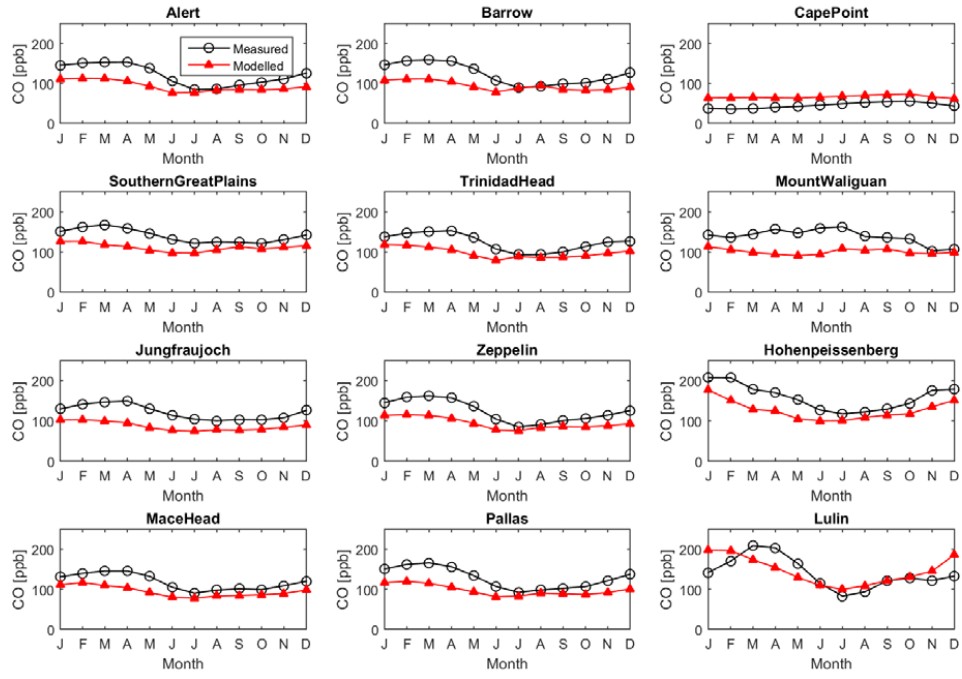


*Figure 3: Monthly measured and modelled surface CO concentration [ppb] averaged over*
*2008-2010.*






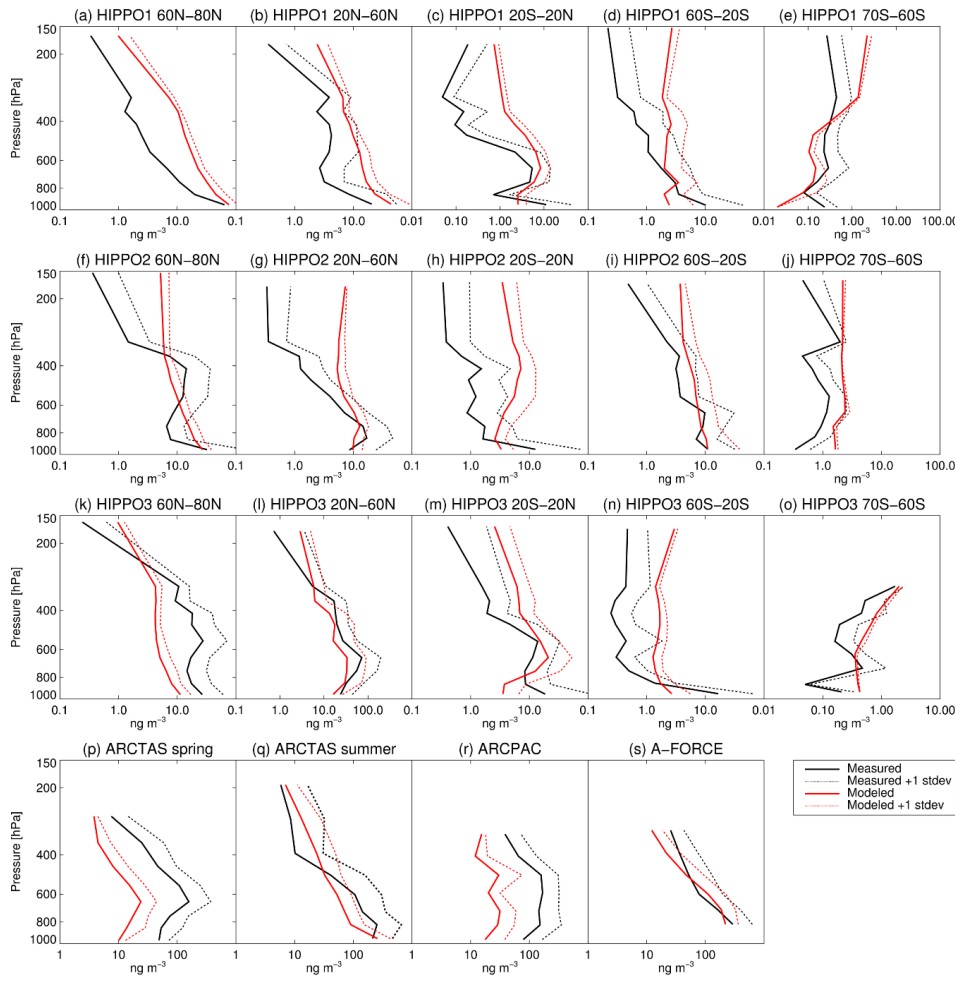


Figure 4: Comparison of modeled vertical profiles of BC with measured rBC from six fligh
campaigns: (a)-(o) HIPPO 1-3, averaged over five latitude bands, (p)-(q) ARCTAS, spring
and summer, (r) ARCPAC and (s) A-FORCE. Solid lines show the average of observations
and model results binned in 100 hPa invervals (25 hPa for HIPPO data between 400 and 200
hPa), while dashed lines show the standard deviation in each interval.




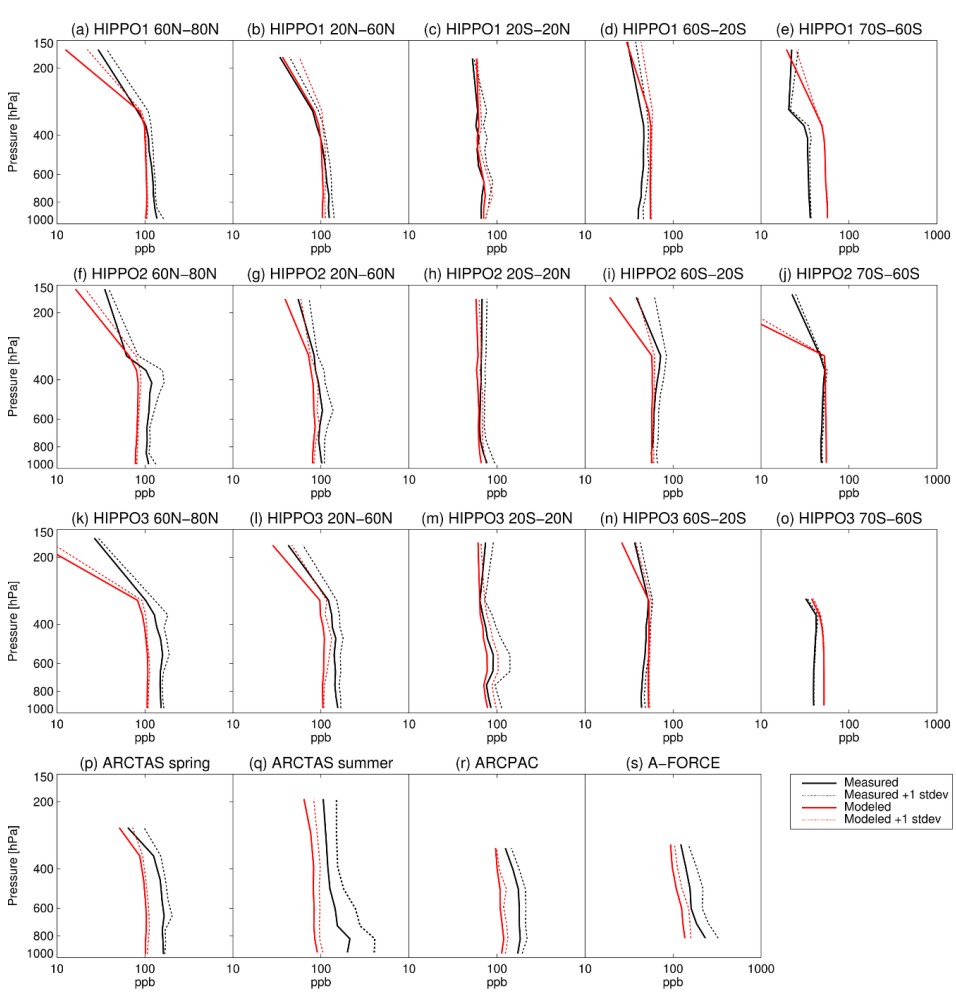


Figure 5: same as Fig. 4, but for CO.






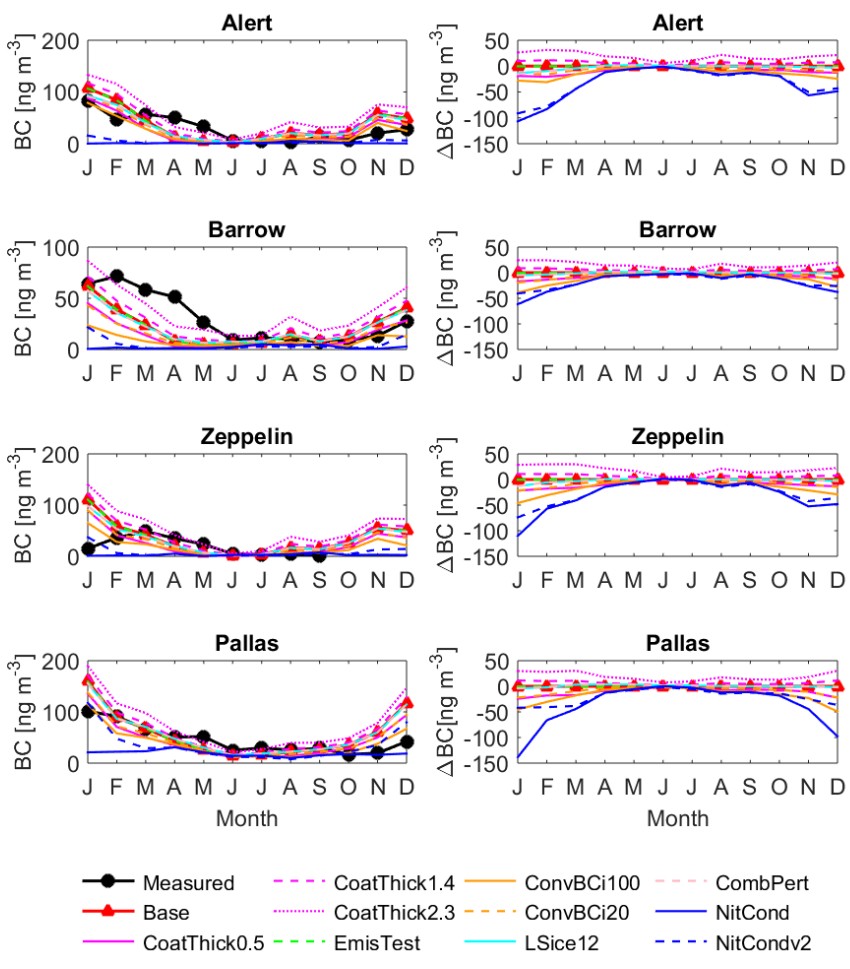


Figure 6: Monthly surface concentrations of BC at Arctic stations in 2008: measurements
versus baseline and sensitivity simulations (right column) and difference between each
sensitivity simulation and the baseline (left).





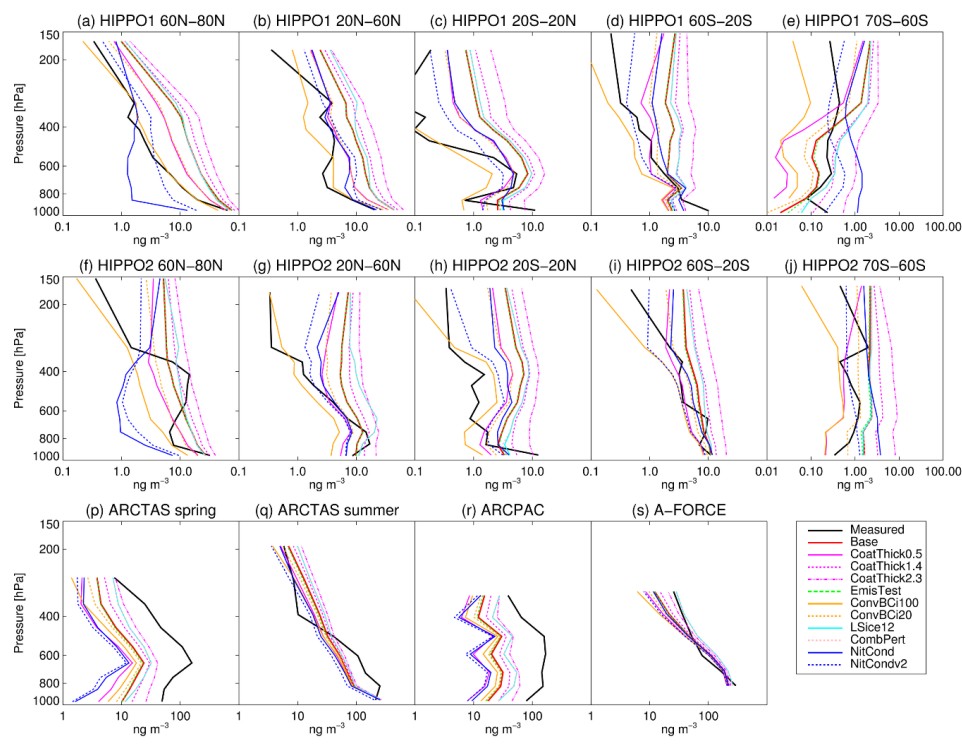


Figure 7: Vertical profiles of BC in the control and sensitivity runs compared to flight
campaigns.





