# Peer review of "Sensitivity of black carbon concentrations and climate impact to aging and scavenging in OsloCTM2-M7"

_Atmospheric Chemistry and Physics, 2016_

## Referee Comment (RC1) · Anonymous Referee #1 · 19 Oct 2016

In this paper, the authors investigated the sensitivity of black carbon (BC) concentrations in the chemistry-transport model OsloCTM2-M7 to parameters controlling aerosol and scavenging. They especially focused on surface concentrations over the Arctic and vertical profiles over remote regions. Many sensitivity simulations were conducted considering the uncertainties in the coating thickness of sulfate, scavenging by convective and ice precipitation, nitrate formation, and emissions, and the authors showed the importance of the BC ice nucleating efficiency and the change in hygroscopicity with aging.

It is very important to understand the sources of uncertainties in simulating BC concentrations especially over remote regions. So, the theme of this paper is interesting and important. However, I feel there are some fundamental problems in the method (the model representation of aging processes) and the description of this paper, as shown

in the major comments below. I suggest the authors to consider these comments carefully because they may be important for the results of this study. The modifications of the model and/or additional sensitivity simulations will be useful to consider these comments.

Major comments:

(1) New findings in this study

What are new findings in this paper scientifically? The authors show the results of many sensitivity simulations, but I think most conclusions obtained from the simulations are already shown by previous studies. For example, previous studies (listed on the references in this paper) showed the uncertainties of BC scavenging by convective precipitation, the poor agreement of BC concentrations over the Pacific (HIPPO) and Arctic (ARCTAS, ARCPAC), the overestimation of BC concentrations at higher altitudes, and sensitivity simulations focused on the aging timescale of BC. Some global aerosol models already consider nitrate formation. Considering these points, I suggest the authors to highlight the important conclusions (new scientific findings) obtained in this study.

(2) BC aging by organic aerosol formation

BC aging processes by organic aerosol (OA) formation will be important because OA mass concentrations are high and are roughly similar to sulfate mass concentrations on global average (at the surface). Considering the concentrations in the atmosphere, OA formation is probably more important than nitrate formation in terms of BC aging by condensation. However, I could not find any description about the BC aging by OA formation. If the OsloCTM2-M7 model does not consider the BC aging by OA formation, the model is insufficient to represent BC aging processes. It is better to improve the model to consider OA formation and the BC aging by OA formation. If it is difficult for the authors to modify the model in a short time, I suggest the authors to add some sensitivity simulations to show the potential uncertainties due to the BC aging by

OA formation processes by using the current model.

(3) BC aging by nitrate

Please clarify the treatment of nitrate evaporation. As for sulfate, it is relatively easy because it is enough to consider the conversion from hydrophobic BC to hydrophilic BC. However, as for nitrate, the conversion of both directions will be important. The evaporation of nitrate is especially important over remote regions, and it may be possible to change from hydrophilic BC to hydrophobic BC over the regions through evaporation of nitrate. If the model already considers the effect of nitrate evaporation, please describe about it and show its importance (e.g., as a sensitivity simulation). If not, please add the effect to the model or please show some results that the effect is not important.

Other comments:

(1) Section 3.1.3

I suggest the authors to add a figure showing the results of this section.

(2) Figures 6 and 7

It is hard to see the lines in Figures 6 and 7. Please revise these figures to make them easy to understand.

(3) The number of ML (lines 491-492)

Please describe why the authors use different MLs for sulfate and nitrate.

(4) Typos

There are some typos in the text. Please correct them.

Line 212: onHoose et al.

Line 581: amont

Line 617: dependen

---

## Short Comment (SC1) · 31 Oct 2016

The authors conducted comprehensive model sensitivity simulations to investigate the impact of BC aging and scavenging on BC concentrations and radiative forcing by evaluating model results with observations over the Arctic and remote Pacific oceans. This study could improve our understanding of how some key processes affect global BC simulation and associated radiative forcing. I have three short comments.

1. For BC aging, the authors included the condensation of sulfuric acid and nitric acid as well as coagulation with sulfate in the model. Recently, He et al. (2016a) implemented a microphysics-based BC aging scheme in a global chemical transport model, which accounts for BC aging due to secondary organic aerosol (SOA) condensation and coagulation in addition to the aging processes in the present study. They showed a

substantial model improvement by comparing with HIPPO observations and that SOA condensation/coagulation also plays an important role in BC aging, particularly over biomass burning source regions. Since the SOA process is not considered here, it would be useful for the authors to add some discussions on this aspect.

Reference:

He, C., Li, Q., Liou, K.-N., Qi, L., Tao, S., and Schwarz, J. P.: Microphysics-based black carbon aging in a global CTM: constraints from HIPPO observations and implications for global black carbon budget, Atmos. Chem. Phys., 16, 3077-3098, doi:10.5194/acp-16-3077-2016, 2016a.

2. For BC simulations over the Arctic, recent study by Qi et al. (2016a) systematically evaluated the effects of model processes such as BC emissions, dry and wet deposition on BC distributions in both the air and the snow surface over the Arctic. They showed that (1) the flaring emissions (natural gas flaring), particularly in North of Russia, has a significant impact on BC concentrations over the Arctic, (2) the low bias in BC dry deposition velocities over snow and ice in current CTMs contributes to model bias in BC simulations, and (3) including the Wegener-Bergeron-Findeisen (WBF) aerosol-cloud process in mixed-phase clouds significantly improves BC simulations both in the Arctic and globally (Qi et al., 2016b). I suggest citing these recent papers and including some discussions on this aspect, since the content is closely related to the analysis and results in the present study.

References:

Qi, L., Li, Q., Li, Y., and He, C.: Factors Controlling Black Carbon Distribution in the Arctic, Atmos. Chem. Phys. Discuss., doi:10.5194/acp-2016-707, in review, 2016a.

Qi, L., Li, Q., He, C., Wang, X., and Huang, J.: Effects of Wegener-Bergeron-Findeisen Process on Global Black Carbon Distribution, Atmos. Chem. Phys. Discuss., doi:10.5194/acp-2016-706, in review, 2016b.

[Figure]

3. For radiative forcing calculation, the authors mentioned that the effects of BC aging on its optical properties is not considered in their model. Recent studies (He et al., 2015, 2016b) showed that BC aging substantially influences its optical properties due to various coating structures and thicknesses, leading to a factor of 2-4 variations in BC absorption and scattering. It would be very helpful if the authors could discuss about this coating effect during aging on BC radiative properties and/or forcing estimates.

References:

He, C., Liou, K.-N., Takano, Y., Zhang, R., Levy Zamora, M., Yang, P., Li, Q., and Leung, L. R.: Variation of the radiative properties during black carbon aging: theoretical and experimental intercomparison, Atmos. Chem. Phys., 15, 11967-11980, doi:10.5194/acp-15-11967-2015, 2015.

He, C., Y. Takano, K. N. Liou, P. Yang, Q. B. Li, and D. W. Mackowski: Intercomparison of the GOS approach, superposition T-matrix method, and laboratory measurements for black carbon optical properties during aging, J. Quant. Spectrosc. Radiat. Transf., 184, 287–296, doi:10.1016/j.jqsrt.2016.08.004, 2016b.

---

## Referee Comment (RC2) · Anonymous Referee #2 · 2 Nov 2016

The paper evaluated simulated black carbon in OsloCTM2-M7 against various observations and performed several sensitivity simulations by varying BC aging and scavenging parameters. The paper particularly focused on improving BC predictions over the high latitude, which is a particularly interesting topic as potentially important role of BC on climate changes occurring in high latitude such as Arctic.

Despite this importance, I have major concerns with this paper. I agree with all the concerns addressed by the referee #1. I particularly agree that this paper does not provide new findings. Here is the list of my major comments. Please consider them to improve this manuscript.

Major comments:

1) Regarding BC modeling in OsloCTM2-M7, please explain any difference/update in BC modeling used in this study compared to the ones used in previous studies.
     Without such information, this paper appears to be very redundant to
     previous studies with OsloCTM2. This is particularly because OsloCTM2 has
     participated several multi-model inter-comparison studies (e.g., AEROCOM)
     focused on black carbon evaluations against observation. Also, there were
     previous studies using OsloCTM2 (maybe with bulk aerosol model) improved
     BC prediction by adjusting aging/deposition parameterization and
     shortening BC lifetime (e.g., Skeie et al. 2011; Hodnebrog et al, 2014).

     The authors should make it clear how the model different from previous
     studies with OsloCTM2, and how BC predictions in this model are improved
     from previous OsloCTM2 evaluation. Specifically, Lund and Berntsen (2012)
     evaluated OsloCTM2-M7 BC predictions against the same observation. Please
     explain how the BC modeling and evaluation results in this paper differ from
     that previous paper.

2) The BC sensitivity results do not seem so informative.
     Large portion of the paper results are focused on BC evaluations, not the
     sensitivity results - This paper actually fit better as OsloCTM2-M7 BC
     evaluation paper, rather than BC sensitivity study. If the authors wish to stay
     focused on BC sensitivity study, I strongly recommend examining details
     comparison (i.e., spatial and temporal distributions of concentrations and
     radiative forcing) among the sensitivity simulations to find any interesting
     spatial and temporal differences. This may be helpful to understand the
     climate impact.

Minor comments:

Abstract section
  1) Please re-write Abstract. I got an impression that the current abstract is just
     a short version of the conclusion section. I found some identical sentences or

phrases between abstract and conclusions. Also, the abstract seems too long and needs to improve readability. Here are some examples:

P1 L12 : please modify "microphysical aerosol to "aerosol microphysical"

P1 L14 : Please clarify "Arctic surface concentrations". Is this BC ambient concentrations or BC in Arctic snow or both?

P1 L14: please modify "remote region BC vertical profiles" to "BC vertical profiles at remote region".

P1 L17 : please modify "annual averaged" to "annually averaged" or "annual average"

P2 L22: Please re-write this sentence: "Several processes can achieve this".

Section 2.3
1) Regarding BC aging by HNO3 condensation, please explain why HNO3 produced in the aq. Chemistry has to be excluded. Is this to estimate gas-phase production?
2) It looks like the required ML is different for sulfate and nitrate. In reality, these hydrophilic aerosols will condense on BC surface and change BC properties. Isn't it more realistic to set the required ML combined for sulfate and nitrate? Am I missing something?

Section 2.4
1) I can't follow the first paragraph describing the method (L226-L238) to distribute BC burden to CESM-CAM4 model. Can you please re-write this method more clearly? Did you have to re-gridding BC burden?

Section 3.1.1
1) L314-L315 : Please provide a citation.
2) L323-340 : This study applies seasonality in agricultural waste burning and domestic BC emissions. What about other emission sources? What is the impact of missing seasonality of other emission sources?
3) L335-336 : Is this for certain year? 2008?
4) L 348-349 : Please present the CO evaluation for SH region.

Section 3.1.2
1) L361 : Please provide a citation.
2) L364-367 : This doesn't apply to ARCTAS summer. Please explain why.
3) L385-386 : I am not sure what this mean. Please explain why it is less important for aerosol distribution.

Conclusion section
1) Please see the comments for Abstract section above, which are also applied to this section as well.
2) L561: put comma between "aging" and "and".
3) L581: please specify how much MNB is changed

4) L584: It looks like this part has a grammatical error: "… available for removal, a parameter with large ".
5) L584 : "uncertaines" typo
6) L587: "fligh" typo
7) L589: please specify how big is the overestimation.
8) L607- L609 : This sentence should be rewritten. It doesn't read well.
9) L610: please change "is" to "are".
10) L617 : please fix this part: "dependen on"
11) L614-618: This is very long sentence and it is not well read. Please re-write this.
12) L618-619: Please explain more what you mean by "tradeoffs … between different regions".
13) L621 –L622: If possible, please specify what kind of observation data that would be especially useful to improve BC modeling?

---

## Author Comment (AC1) · 5 Jan 2017

**Response to review of "Sensitivity of black carbon concentrations and climate impact to aging and scavenging" by Marianne T. Lund, Terje K. Berntsen and Bjørn H. Samset.**

We thank the anonymous referee for the carefully and thorough review of our paper and the useful suggestions. Responses to individual comments are given below.

**Anonymous Referee #1**

In this paper, the authors investigated the sensitivity of black carbon (BC) concentrations in the chemistry-transport model OsloCTM2-M7 to parameters controlling aerosol and scavenging. They especially focused on surface concentrations over the Arctic and vertical profiles over remote regions. Many sensitivity simulations were conducted considering the uncertainties in the coating thickness of sulfate, scavenging by convective and ice precipitation, nitrate formation, and emissions, and the authors showed the importance of the BC ice nucleating efficiency and the change in hygroscopicity with aging.

It is very important to understand the sources of uncertainties in simulating BC concentrations especially over remote regions. So, the theme of this paper is interesting and important. However, I feel there are some fundamental problems in the method (the model representation of aging processes) and the description of this paper, as shown in the major comments below. I suggest the authors to consider these comments carefully because they may be important for the results of this study. The modifications of the model and/or additional sensitivity simulations will be useful to consider these comments.

Major comments:
(1) New findings in this study
What are new findings in this paper scientifically? The authors show the results of many sensitivity simulations, but I think most conclusions obtained from the simulations are already shown by previous studies. For example, previous studies (listed on the references in this paper) showed the uncertainties of BC scavenging by convective precipitation, the poor agreement of BC concentrations over the Pacific (HIPPO) and Arctic (ARCTAS, ARCPAC), the overestimation of BC concentrations at higher altitudes, and sensitivity simulations focused on the aging timescale of BC. Some global aerosol models already consider nitrate formation. Considering these points, I suggest the authors to highlight the important conclusions (new scientific findings) obtained in this study.

The objective of our study is to explore the range of results under varying assumptions in a specific model, how these influence existing model-measurement discrepancies and identify potential improvements that can be implemented before further applications of this model. This is crucial in order to advance BC modelling, e.g. as several recent studies have documented that the current model ensembles do not accurately reproduce measured BC vertical profiles. In the years to come, several new aircraft campaigns are planned. It is of imperative that the modelling groups carefully document the current performance of the global models, before further comparison against new measurements. Furthermore, information about the sensitivity of BC to key processes and parameters may contribute insight to where efforts could be focused in upcoming campaigns in order to provide the best possible data for further constraining global models. Since the global

models differ considerably in their treatment of aerosols aging and scavenging, it is important to examine a broad range of processes in a several models.

However, we also go beyond testing of model performance, to ensure that our results contribute to the growing body of literature on BC modeling. E.g., we focus simultaneously on model capabilities at high latitudes and remote regions over the Pacific, whereas previous studies often focus on one or the other. Additionally, using a microphysical module allows us to investigate parameters beyond those examined in studies using bulk modules (e.g., Hodnebrog et al. (2014)), thereby providing additional information about the importance of underlying processes. Finally, as input to the discussion surrounding the role of BC in the climate system, we also move beyond differences in concentrations and examine the consequent impact on global BC radiative forcing and temperature response.

To better reflect our main objective and the points above, we have changed the title and modified the abstract, introduction and conclusions sections.

(2) BC aging by organic aerosol formation

BC aging processes by organic aerosol (OA) formation will be important because OA mass concentrations are high and are roughly similar to sulfate mass concentrations on global average (at the surface). Considering the concentrations in the atmosphere, OA formation is probably more important than nitrate formation in terms of BC aging by condensation. However, I could not find any description about the BC aging by OA formation. If the OsloCTM2-M7 model does not consider the BC aging by OA formation, the model is insufficient to represent BC aging processes. It is better to improve the model to consider OA formation and the BC aging by OA formation. If it is difficult for the authors to modify the model in a short time, I suggest the authors to add some sensitivity simulations to show the potential uncertainties due to the BC aging by OA formation processes by using the current model.

The M7 accounts for interaction with organic carbon through coagulation, but is so far only limited to primary organic carbon and does not include condensation by secondary organics. While the OsloCTM2-M7 includes a treatments for the gas-aerosol partitioning of secondary organics, a coupling of this module with the M7 require resources and time beyond that is available for this study. Furthermore, the objective of the current study is not to develop a new aerosol parameterization, but to test the range of concentrations and vertical profiles to changes in selected parameters. However, we agree that the potential limitations of not accounting for secondary organics should be made clear and have added the following paragraph:

*"In addition to nitrate, condensation of organic aerosols could play an important role in the aging of BC. For instance, He et al. (2016) recently found that a microphysics-based BC aging scheme including condensation of both nitric acid and secondary organics resulted in improved representation of BC in GEOS-Chem compared with HIPPO measurements. This process is currently not included in the OsloCTM2-M7, but should be addressed in future work."*

(3) BC aging by nitrate

Please clarify the treatment of nitrate evaporation. As for sulfate, it is relatively easy because it is enough to consider the conversion from hydrophobic BC to hydrophilic BC. However, as for nitrate, the conversion of both directions will be important. The evaporation of nitrate is especially important over remote regions, and it may be possible to change from hydrophilic BC to hydrophobic BC over the regions through evaporation of nitrate. If the model already considers the effect of nitrate evaporation, please describe about it and show its importance (e.g., as a

sensitivity simulation). If not, please add the effect to the model or please show some results that the effect is not important.

The referee raises an important point. Changes in BC hydrophilicity due to evaporation of nitric acid is not something we have considered in our simulations. In this way, our sensitivity test likely represent an upper estimate of the efficiency of nitric acid in the BC aging process. Furthermore, we find very little literature on the parameterization of this process or its impact. To highlight the uncertainty and limitation in our study, we have added the following paragraph:

*"Another important caveat is that we do not account for changes in hydrophilicity resulting from evaporation of nitric acid already condensed on the aerosols. This may result in an overestimation of the contribution from nitric acid to the aging, at least in certain regions."*

Other comments:
(1) Section 3.1.3
I suggest the authors to add a figure showing the results of this section.

Based on the comment by referee #2 regarding the balance of the paper, we have chosen to focus more on the sensitivity studies and somewhat less on the evaluation. We have therefore chosen to include a shortened version of the BC in snow documentation in the section describing the surface concentrations, rather than as a separate section. In light of this, we also do not include additional figures of model evaluation.

(2) Figures 6 and 7
It is hard to see the lines in Figures 6 and 7. Please revise these figures to make them easy to understand.

The figures have been revised with changes to the colors and line thickness.

(3) The number of ML (lines 491-492)
Please describe why the authors use different MLs for sulfate and nitrate.

We realize this description is very unclear. There is only one variable giving the number of MLs required for moving a BC aerosol from the insoluble to the soluble mode and the number of particles than can be moved, i.e., has sufficient associated soluble material, is determined from the total sulfate and nitric acid condensation. However, when adding nitric acid, we perform additional simulations with 5 and 10 MLs, reflecting the range of values used in previous studies. The text has been clarified:

*"The number of MLs used as the criterion for aging ranges in existing literature. In its original setup M7 assumes 1 ML, based on the best agreement with a sectional model found by Vignati et al. (2004), but this consider sulfate as the only condensable species. Other studies have used a 5 (Pringle et al., 2010) and 10 (Mann et al., 2010) monolayer scheme. Reflecting this range and examining the subsequent impact on concentrations, we here perform three runs assuming 1, 5 and 10 ML are required for aging."*

(4) Typos
There are some typos in the text. Please correct them.
Line 212: onHoose et al.
Line 581: amont
Line 617: dependen

Typos have been corrected.

---

## Author Comment (AC2) · 5 Jan 2017

We thank Dr. Cenlin He for useful comments and for making us aware of the additional literature.

1) We agree and have added a paragraph discussing the limitation of not considering secondary organic aerosols in the aerosol microphysics parameterization in our study, including reference to He et al. (2016a).

2) These are indeed important issues. As for i), we already include this topic. Regarding ii), we have added a comparison of our study with the paper by Fan et al. (2012) who investigate ice formation processes, including the WBF process (we do not cite the Qi et al. (2016a) paper at this point as it is still under review).

[Figure]

**[ACPD](# )**

Interactive
comment

3) The absorption enhancement is a key factor for the RF of BC. In our RF calculations, we use the precalculated kernel from Samset and Myhre (2015). Samset and Myhre (2015) discussed the missing coating effect during BC aging in CAM4 in detail. We have added a reference to that discussion, but do not include further details our study since we focus on RF differences from the baseline case.

---

## Author Comment (AC3) · 5 Jan 2017

**Response to review of "Sensitivity of black carbon concentrations and climate impact to aging and scavenging" by Marianne T. Lund, Terje K. Berntsen and Bjørn H. Samset.**

We thank the anonymous referee for the carefully and thorough reviews of our paper and the useful suggestions. Responses to individual comments are given below.

**Anonymous Referee #2**

The paper evaluated simulated black carbon in OsloCTM2-M7 against various observations and performed several sensitivity simulations by varying BC aging and scavenging parameters. The paper particularly focused on improving BC predictions over the high latitude, which is a particularly interesting topic as potentially important role of BC on climate changes occurring in high latitude such as Arctic.
Despite this importance, I have major concerns with this paper. I agree with all the concerns addressed by the referee #1. I particularly agree that this paper does not provide new findings. Here is the list of my major comments. Please consider them to improve this manuscript.

Major comments:
1) Regarding BC modeling in OsloCTM2-M7, please explain any difference/update in BC modeling used in this study compared to the ones used in previous studies. Without such information, this paper appears to be very redundant to previous studies with OsloCTM2. This is particularly because OsloCTM2 has participated several multi-model inter-comparison studies (e.g., AEROCOM) focused on black carbon evaluations against observation. Also, there were previous studies using OsloCTM2 (maybe with bulk aerosol model) improved BC prediction by adjusting aging/deposition parameterization and shortening BC lifetime (e.g., Skeie et al. 2011; Hodnebrog et al, 2014). The authors should make it clear how the model different from previous studies with OsloCTM2, and how BC predictions in this model are improved from previous OsloCTM2 evaluation. Specifically, Lund and Berntsen (2012) evaluated OsloCTM2-M7 BC predictions against the same observation. Please explain how the BC modeling and evaluation results in this paper differ from that previous paper.
Lund and Berntsen (2012) performed the first analysis of BC simulated by the M7 in the OsloCTM2 and compared the M7 with the standard bulk parameterization (OsloCTM2-BULK). A basic evaluation against selected measurements was performed, showing that the M7 improved the representation of Arctic surface concentrations compared with the bulk scheme, but that considerable overestimate of high altitude concentrations remained a problem. Because the M7 significantly increases the required computing time, the simulations of Lund and Berntsen (2012) were used to generate a latitudinally and seasonally varying look-up table of aging rates for use in the bulk scheme as documented in Skeie et al. (2011). In this way, at least some of the spatial and temporal variability in aging could be accounted for in the bulk scheme. However, this approach can only capture variations under present-day conditions, e.g., changes in variability with changing emissions not captures. Furthermore, despite existing issues, using an aerosol microphysical module like M7 provides a physically more realistic parameterization, which we believe is needed in order to improve the modeling of aerosols. Building on the findings in Lund and Berntsen (2012), we here perform a much more thorough documentation of the model (which is also needed due to

recent important updates to the emission inventories) and explore possible reasons for the high-altitude discrepancies identified previously. Moreover, using a microphysical module allows us to investigate parameters beyond those examined by Hodnebrog et al. (2014) who used the bulk aerosol module. We also provide additional information by focusing simultaneously on model capabilities at high latitudes and remote regions over the Pacific; whereas other sensitivity studies often focus on one or the other. See also response to comment 1) by referee #1.

To clarify, we have modified the introduction section:

*"Here we examine the sensitivity of modeled BC concentrations to factors controlling aerosol lifetime in the OsloCTM2 (Sovde et al., 2008) coupled with the aerosol microphysical parameterization M7 (Vignati et al., 2004) (hereafter OsloCTM2-M7). The chemical transport model OsloCTM2 has been documented and used in several multi-model aerosol studies (Balkanski et al., 2010; Myhre et al., 2013; Schulz et al., 2006; Shindell et al., 2013; Textor et al., 2007). These studies used a simplified bulk aerosol scheme. Lund and Berntsen (2012) (hereafter LB12) performed the first analysis of BC simulated by the M7 in the OsloCTM2 and compared results with those from the bulk parameterization. A basic evaluation against selected measurements was performed, showing that using M7 improved the representation of Arctic surface concentrations compared with the bulk scheme, but exacerbated the overestimation of high-altitude BC.*

*Building on the findings in LB12, we perform a range of sensitivity experiments varying key assumptions in the treatment of aging and scavenging in OsloCTM2-M7 and investigate the resulting range in vertical BC profiles, as well as high-latitude surface concentrations. Using updated emission inventories, three years of model results and observations from surface stations, flight campaigns, and snow samples, we also perform a more thorough documentation of the current model performance. Our experiments include a first step towards accounting for BC aging by gas-phase nitric acid condensation. Measurements have shown that nitrate is frequently present in internal aerosol mixtures (Pratt & Prather, 2010; Shiraiwa et al., 2007). Aging through interaction with nitrate may also become more important in the future following strong projected decreases in $SO_2$ emissions and increasing $NOx$ and greenhouse gas emissions (Bauer et al., 2007; Bellouin et al., 2011; Makkonen et al., 2012), but has so far not been accounted for in the model. We also take the analysis one step further and estimate the range in global RF and surface temperature resulting from the changes in the model parameters.."*

2) The BC sensitivity results do not seem so informative. Large portion of the paper results are focused on BC evaluations, not the sensitivity results - This paper actually fit better as OsloCTM2-M7 BC evaluation paper, rather than BC sensitivity study. If the authors wish to stay focused on BC sensitivity study, I strongly recommend examining details comparison (i.e., spatial and temporal distributions of concentrations and radiative forcing) among the sensitivity simulations to find any interesting spatial and temporal differences. This may be helpful to understand the climate impact.

This is a good point. While we would argue that a careful documentation of the model performance in its original setup is needed before examining the range of results in the sensitivity tests and attempting to identify potential improvements, we agree with the referee that the paper should increase the focus on the sensitivity experiments. We have kept most of text describing the model evaluation, but shortened it where possible. We have also moved the figures showing the model-measurement comparison of other species than BC to a supplementary material. We also analyze

the difference in spatial distribution of BC concentrations in the sensitivity experiments in more detail, including two new figures. Finally, the global vertical profiles of forcing and temperature response are examined in more detail.

Minor comments:
Abstract section
1) Please re-write Abstract. I got an impression that the current abstract is just a short version of the conclusion section. I found some identical sentences or phrases between abstract and conclusions. Also, the abstract seems too long and needs to improve readability. Here are some examples:
P1 L12 : please modify "microphysical aerosol to "aerosol microphysical"
P1 L14 : Please clarify "Arctic surface concentrations". Is this BC ambient concentrations or BC in Arctic snow or both?
P1 L14: please modify "remote region BC vertical profiles" to "BC vertical profiles at remote region".
P1 L17 : please modify "annual averaged" to "annually averaged" or "annual average"
P2 L22: Please re-write this sentence: "Several processes can achieve this".
The abstract has been rewritten and minor comments below addressed where still applicable:
*"Accurate representation of black carbon (BC) concentrations in climate models is a key prerequisite for understanding its net climate impact. BC aging scavenging are treated very differently in present models. Here, we examine the sensitivity of 3-dimensional, temporally resolved BC concentrations to perturbations to individual model processes in the chemistry-transport model OsloCTM2-M7. The main goals are to identify processes related to aerosol aging and scavenging where additional observational constraints may most effectively improve model performance, in particular for BC vertical profiles, and to give an indication of how model uncertainties in the BC life cycle propagate into uncertainties in climate impacts. Coupling OsloCTM2 with the microphysical aerosol module M7 allows us to investigate aging processes in more detail than possible with a simpler bulk parameterization. Here we include, for the first time in this model, a treatment of condensation of nitric acid on BC. Using radiative kernels, we also estimate the range of radiative forcing and global surface temperature responses that may result from perturbations to key tunable parameters in the model. We find that BC concentrations in OsloCTM2-M7 are particularly sensitive to convective scavenging and the inclusion of condensation by nitric acid. The largest changes are found at higher altitudes around the Equator and at low altitudes over the Arctic. Convective scavenging of hydrophobic BC, and the amount of sulfate required for BC aging, are found to be key parameters, potentially reducing bias against HIPPO flight-based measurements by 60 to 90 percent. Even for extensive tuning, however, the total impact on global mean surface temperature is estimated to less than 0.04K. Similar results are found when nitric acid is allowed to condense on the BC aerosols. We conclude, in line with previous studies, that a shorter atmospheric BC lifetime broadly improves the comparison with measurements over the Pacific. However, we also find that the model-measurement discrepancies can not be uniquely attributed to uncertainties in a single process or parameter. Model development therefore needs to be focused on improvements to individual processes, supported by a broad range of observational and experimental data, rather than tuning of individual, effective parameters such as the global BC lifetime."*

Section 2.3
1) Regarding BC aging by HNO3 condensation, please explain why HNO3 produced in the aq. Chemistry has to be excluded. Is this to estimate gasphase production?
Yes, this is to estimate gas-phase production of HNO3. We realize that since it's in fact the gas-phase nitric acid we're after here, showing results also from the case including the aqueous phase production is confusing and not very informative. We have therefore removed the latter experiment in order to focus on the correct one.

2) It looks like the required ML is different for sulfate and nitrate. In reality, these hydrophilic aerosols will condense on BC surface and change BC properties. Isn't it more realistic to set the required ML combined for sulfate and nitrate? Am I missing something?
We realize this description is very unclear (this was also pointed out by referee #1). There is only one variable giving the number of MLs required for moving a BC aerosol from the insoluble to the soluble mode and the number of particles than can be moved, i.e., has sufficient associated soluble material, is determined from the total sulfate and nitric acid condensation. However, when adding nitric acid, we perform additional simulations with 5 and 10 MLs, reflecting the range of values used in previous studies. The text has been clarified:
*"The number of MLs used as the criterion for aging ranges in existing literature. In its original setup M7 assumes 1 ML, based on the best agreement with a sectional model found by Vignati et al. (2004), but this considers sulfate as the only condensable species. Other studies have used a 5 (Pringle et al., 2010) and 10 (Mann et al., 2010) monolayer scheme. Reflecting this range and examining the subsequent impact on concentrations, we here perform three runs assuming 1, 5 and 10 ML are required for aging."*

Section 2.4
1) I can't follow the first paragraph describing the method (L226-L238) to distribute BC burden to CESM-CAM4 model. Can you please re-write this method more clearly? Did you have to re-gridding BC burden?
We have expanded and clarified this method description and the section now reads:
*"To estimate implications of the concentration changes in our experiments for the global BC climate impact, we use precalculated radiative forcing (RF) and surface temperature (TS) kernels derived with the CESM-CAM4 (Samset & Myhre, 2015). These 3-dimensional, temporally varying kernels were constructed by systematically applying a uniform BC burden to one model layer at a time, and calculating the resulting responses. Effective radiative forcing (ERF) was extracted from simulations with prescribed sea-surface temperatures, while temperature responses were taken from simulations with a slab ocean setup. As shown in (Samset & Myhre, 2015), it is possible to take a perturbation to the 3D concentration, multiply it with the kernels, and get an estimate for the resulting ERF and temperature change. However, because the BC perturbations were applied uniformly throughout a single model layer, the temperature response at each grid point will be due to both BC forcing exerted locally and to forcing in surrounding gridboxes. In the present analysis, we therefore focus on global mean vertical profiles. For each experiment, the globally averaged vertical BC profile from the OsloCTM2-M7 is multiplied with the globally averaged vertical forcing and temperature change kernels, respectively. The kernels are interpolated to the OsloCTM2-M7 resolution before use."*

Section 3.1.1
1) L314-L315 : Please provide a citation.
Text slightly modified and citation added:
*"Studies have found that models often struggle to capture the seasonal cycle and magnitude of measured high-latitude BC surface concentrations (e.g., Eckhardt et al. 2015; Shindell et al. 2008)."*

2) L323-340 : This study applies seasonality in agricultural waste burning and domestic BC emissions. What about other emission sources? What is the impact of missing seasonality of other emission sources?
The ECLIPSEv4 emission inventory used in this study does not include seasonality of other sectors than domestic and agricultural waste burning (AWB). However, monthly data is provided in a more recently released version, ECLIPSEv5. Aside from AWB and domestic emissions, the seasonal variation in this inventory is negligible, both globally and north of 30 degree north. We have added the following text in the methods section 2.2:
*"Seasonality of emissions in other sectors is not included in ECLIPSEv4. In the more recently released ECLIPSEv5 inventory (MLhttp://eclipse.nilu.no/, the monthly variability in emissions from other sectors is minor or negligible."*

3) L335-336 : Is this for certain year? 2008?
The emissions in the Wiedinmyer et al. (2014) study is based on year 2010 population and waste generation data. This has been clarified in the manuscript.

4) L 348-349 : Please present the CO evaluation for SH region.
We have included the SH evaluation in the supplementary material (see response to comment major comment #2 above).

Section 3.1.2
1) L361 : Please provide a citation.
Citations have been added and the text modified as follows for clarification:
*"During April 2008, when these campaigns were undertaken, there was unusually strong fire activity in Siberia and air masses were heavily influenced by biomass burning emissions (Brock et al., 2011; Jacob et al., 2010; Warneke et al., 2009). During several flights, the biomass burning plumes were specifically targeted. Possible reasons the strong discrepancies could be underestimation of the fire emissions during these extreme fire events or inaccurate representation of the plumes in the model."*
Jacob, D. J., et al.: The Arctic Research of the Composition of the Troposphere from Aircraft and Satellites (ARCTAS) mission: design, execution, and first results, Atmos. Chem. Phys., 10, 5191-5212, doi:10.5194/acp-10-5191-2010, 2010
Brock, C. A., et al.: Characteristics, sources, and transport of aerosols measured in spring 2008 during the aerosol, radiation, and cloud processes affecting Arctic Climate (ARCPAC) Project, Atmos. Chem. Phys., 11, 2423-2453, doi:10.5194/acp-11-2423-2011, 2011.
Warneke, C., *et al. (2009),* Biomass burning in Siberia and Kazakhstan as an important source for haze over the Alaskan Arctic in April 2008, Geophys. Res. Lett., *36, L02813*

2) L364-367: This doesn't apply to ARCTAS summer. Please explain why.

The majority of flights during the summer campaign of the ARCTAS took place over Canada, further south than the spring campaign. During summer 2008, there was considerable fire activity in Northern California and Siberia, but the fire activity over Canada was generally low. One possible reason for the better agreement with the measurements from ARCTAS summer is that the model was better able to reproduce the plume transport in this region and/or time of year. Our evaluation against monthly surface concentrations of BC also suggest a generally better agreement at high latitudes during summer than spring. Of course, these differences also underline that flight campaigns only provide a snap shot comparison and that one should be careful not to generalize results. The following paragraph has been added:

*"The majority of flights during the ARCTAS summer campaign took place over Canada, where the fire activity was generally low that year. Moreover, our evaluation against monthly surface concentrations of BC also suggest a generally better agreement at high latitudes during summer than spring (Sect. 3.1.1)."*

3) L385-386 : I am not sure what this mean. Please explain why it is less important for aerosol distribution.

We agree that this is unclear. The sentence in question has been deleted and the following sentence modified to:

*"In contrast, the HIPPO campaigns sampled older air masses, where loss processes have had more time to influence the distribution."*

Conclusion section

1) Please see the comments for Abstract section above, which are also applied to this section as well.

We have rewritten the conclusion section to the comments above; please see revised manuscript.

2) L561: put comma between "aging" and "and".

Corrected

3) L581: please specify how much MNB is changed

Specification has been added and text slightly modified accordingly:

*"Allowing for convective scavenging of hydrophobic BC and reducing the amount of soluble material required for aging results in a 60 to 90 percent lower MNB in the comparison with vertical profiles from HIPPO, relative to the baseline."*

4) L584: It looks like this part has a grammatical error: "… available for removal, a parameter with large ".

Corrected

5) L584 : "uncertaines" typo

Corrected

6) L587: "fligh" typo

Corrected

7) L589: please specify how big is the overestimation.

No longer applicable after this section has been rewritten.

8) L607- L609 : This sentence should be rewritten. It doesn't read well.

We agree and have modified the sentence:

*"In the experiments resulting the most pronounced BC concentration changes relative to the baseline, we calculate changes in global RFari (i.e., direct RF) on the order of 10-30% of the total pre-industrial to present BC direct forcing."*

9) L610: please change "is" to "are".
Corrected
10) L617 : please fix this part: "dependen on"
Corrected
11) L614-618: This is very long sentence and it is not well read. Please re-write this.
We agree and have rephrased:
*"The existing model-measurement discrepancies in the OsloCTM2-M7 can not be uniquely attributed to uncertainties in a single process or parameter."*
12) L618-619: Please explain more what you mean by "tradeoffs … between different regions".
Modified:
*"Furthermore, improvements compared to measurements in one geographical region, can be accompanied by a poorer model performance in other."*
13) L621 –L622: If possible, please specify what kind of observation data that would be especially useful to improve BC modeling?
Examples have been added: *"(…) e.g., of BC IN efficiency, aging rate and mixing state (…)"*

---

## Author Response (AR2)

**Response to review of "Sensitivity of black carbon concentrations and climate impact to aging and scavenging in OsloCTM2-M7" by Marianne T. Lund, Terje K. Berntsen and Bjørn H. Samset.**

We thank the anonymous referee for the carefully and thorough review of our paper and the useful suggestions. Responses to individual comments are given below.

**Anonymous referee#3**

The paper represents a useful addition to the growing body of literature concerning the sensitivity of BC concentrations to parameterization uncertainties. The paper would benefit from a more detailed description of the modeling approach and discussion of key results. In particular, the meaning of results from sensitivity tests with modified scavenging of hydrophobic BC (ConvBC) and condensation of nitrate (NitCond) is not sufficiently clear.

Major comments:

How much does convective aerosol scavenging contribute to total deposition? How is convective scavenging and transport represented in the model? Is a mass flux scheme used and how are aerosols treated in that scheme? Are interactions between entrainment, detrainment, and scavenging processes in convective aerosol mass budgets accounted for?
We have added a description of the parameterization of large-scale and convective cloud systems, to compliment the description of assumptions related to BC solubility and removal. The following paragraph has been included:

*"Wet deposition is calculated based on ECMWF data for convective activity, cloud fraction and rain fall, and on the solubility of individual species. Removal in large-scale cloud systems follow the scheme by Berge [1993]. The parameterization of deep convection is based on the Tiedtke mass flux scheme [Tiedtke, 1989], with mass redistributed in the vertical by a so-called "elevator" principle, i.e., surplus or deficit of mass in the column [Berglen et al., 2004]."*

Why do BC concentrations respond strongly to changes in the scavenging of hydrophobic BC in the convection? Various studies have shown that externally mixed BC particles are very rare in the remote atmosphere which implies that concentrations of hydrophobic BC should be quite low. This would likely imply a low sensitivity of model results to hydrophobic aerosol concentrations in model sensitivity studies. Please quantify the amount of hydrophobic BC in the model and further information about aging time scales etc.

This is an important observation. In fact, in the OsloCTM2-M7 baseline setup, the insoluble Aitken model BC particles (BCi) make up the largest fraction of total BC at both high latitudes and around the tropics. The high-latitude BCi concentrations were also noted by Lund and Berntsen (2012), but we agree that a discussion of this feature is needed in the present study as well. We have added a figure showing the zonal mean concentration of total BC in the baseline simulation and the relative contribution of BCi to the total in the supplementary material. The output required to

calculate aging time scales for all sensitivity experiments (i.e., separate information of the wet removal of BCi) is not available, but we quantify the aging time in the baseline case. The following has been added to the discussion:

*"The strong sensitivity of concentrations at high northern latitudes and around the tropics to changes in the convective scavenging of hydrophobic BC and inclusion of aging by nitric acid may seem surprising given that measurement suggest that significant the majority of freshly emitted hydrophobic BC particles quickly acquire coatings and become internally mixed (Gong et al., 2016; McMeeking et al., 2011; Moteki et al., 2007). Consequently, externally mixed BC particles are likely rare in the remote atmosphere. However, there is still little information about the aerosol mixing state in aged air masses, especially at high latitudes (Raatikainen et al., 2015). In the baseline OsloCTM2-M7, a considerable fraction of total BC is in the insoluble Aitken mode BC (BCi) (45%, or 60 Gg, of the global BC burden with an aging time scale of 2.7 days). In particular, BCi constitutes the dominating fraction of total annual mean BC concentration north and south of 60° (Fig. SI4 b). However, there are important seasonal variations and high-latitude BCi concentration is highest during winter, when the aging is slower due to less efficient production of sulfate. Convective scavenging can be an important loss mechanism during Arctic winter, resulting in a considerable reduction in BC concentrations in the ConvBCi experiment. BCi also dominates above 300 hPa around the tropics."*

What sources and sinks of nitrate are accounted for in the model? Do simulated aerosol size distribution respond to changes in nitrate mass and how is this response parameterized? Is ammonia accounted for in the calculation of the thermodynamic nitrate/HNO3 equilibrium and how is ammonia represented in the model?

The description of EQSAM in Sect. 2.1 has been expanded to include the role and source of ammonia. In response to comment #1 above, a general description of wet and dry deposition in the model is included. As already mentioned, the source of nitric acid is photochemistry. We have modified slightly to clarify and included a reference for further details. A more detailed description of EQSAM is beyond the scope of the study and we refer to further details in the cited literature.

*"Based on the ammonium to sulfate ratio, EQSAM first calculates the preferred state of sulfate. Excess ammonia is available to partition to the aerosol phase, together with gaseous nitric acid, as described in Myhre et al. (2006). Emissions of ammonia are described in Sect. 2.2, while nitric acid is produced through photochemistry as described in Berntsen and Isaksen (1997)."*

We have also clarified that ammonium-nitrate aerosols are not included in M7 and that nitric acid condensation only contributes to transfer of BC from the insoluble to soluble Aitken mode. The following is added in Sect. 2.3:

*"In this experiment, nitric acid contributes only to the transfer of BC from insoluble to soluble Aitken mode, with no further impact on aerosol size distribution."*

Results from the study seem to differ from results of at least 2 other models in the Arctic (Mahmood et al., 2016). In these models, convective scavenging of BC mainly affects BC in the upper troposphere and lower stratosphere, which seems opposite to responses shown in Fig. 3 and 4. Why are concentrations near the surface sensitive to convective scavenging, which presumably occurs outside the Arctic? This does not seem plausible considering that transport of BC to the Arctic is mainly isentropic. Similar, aging processes are more likely to affect mid and upper tropospheric concentrations of BC in the Arctic than near-surface concentrations. It would be beneficial to change units to ppb in the graphs in order to illustrate impacts of scavenging processes relative to transport processes in the simulations.

As described in response to comment #2 above, the model has a high fraction BC mass residing in the insoluble Aitken mode, in particular in the lower Arctic atmosphere during winter, when convective precipitation connected to cold fronts can be an important loss mechanism. Including convective scavenging of hydrophobic BC therefore gives a large impact on the concentrations in this region.

We agree that changing the units to ppb would give a better picture of the influence of transport and scavenging processes. When doing this, there is a general agreement with the Mahmood et al. (2016) with the largest change occurring at around 200 hPa, thereby illustrating the importance of convective scavenging at higher altitudes. We show this here for our ConvBC experiment.

[Figure]

There is, however, little BC at these altitudes. On the other hand, the changes in the BC mass concentration resulting from the changes to the model's parameterizations are important for the consequent impact on climate. For instance, the climate impact is stronger for near-surface BC, especially in the Arctic. To better be able to illustrate the potential climate impact of changes in our experiments as discussed in later sections of the paper, we therefore consider it beneficial to keep the current unit.

A mean normalized bias (MNB) is introduced but is only applied to a few select simulation results, which limits the usefulness of this metric. In addition, the root mean square error, or similar metric, should be considered, too.

We have added MNB for vertical profiles of CO and replaced the correlation coefficient in Fig. S1 with the RMSE.

The main purpose of including a metric for model performance is to quantify potential improvements in model-measurement comparison in the experiments. We feel that the MNB serves this purpose.

Further comments:

P 2 L 62: Allen and Landuyt (2014) seems relevant in this context, too.

Yes. The reference has been added.

P 4 L 112: Is below-cloud scavenging accounted for?

Below cloud scavenging of BC is not included. This has been specified.

P 5 L 159: How is soluble material formed in the model? Through condensation?

As described in the preceding paragraphs of Section 2.1, the sulfate that contributes to aging of insoluble particles is produced through gas-phase oxidation of $SO_2$ by OH. To clarify we have added:

*"(…) (i.e., sulfate from gas-phase oxidation of $SO_2$) (…)"*

P 7 L 224: 3D kernels are apparently available but global mean concentration profiles are used to calculate radiative forcings. Why? A lack of regional information limits the analysis for the Arctic in the second half of the paper.
In the model simulations used to construct the forcing and temperature kernels, the BC perturbations were applied uniformly throughout a single model layer. Consequently, the temperature response at each grid point will be due to BC forcing exerted both locally and in surrounding grid boxes. At a given grid point, the temperature response cannot be uniquely attributed to the concentration change or forcing in that grid box and this kernel-based approach is less useful for a detailed grid point analysis. In addition, due to the significant altitudinal dependence of forcing and temperature response on the BC perturbation, we are mainly interested in the impact of changes in the vertical here.

For radiative forcing, using the full 3D kernels is justified; here the spatially averaged profiles are used simply for consistency with the temperature response calculations. We have confirmed that the prior averaging of the profiles has a <10% impact on the net RF results presented, except in two cases where differences are approx. 20%. For a fuller discussion on this issue, we refer to Stjern et al. (ACP, 2016).

However, note that we do not only use globally averaged profiles. In additional to quantifying the global mean RF and temperature response to global mean concentration changes, the Arctic changes are estimated using Arctic averaged vertical concentrations. We think the description of the methodology is perhaps a bit unclear here and have tried to clarify further. The description in Sect. 2.4 now reads:

*"Combined with the strong altitudinal dependence of forcing and temperature response on the BC perturbation, we therefore focus on horizontally averaged vertical profiles and not on changes at the grid point level in the present analysis. For each experiment, the globally averaged vertical BC profile from the OsloCTM2-M7 is multiplied with the globally averaged vertical forcing and temperature change kernels, respectively. The prior averaging of the profiles has a small impact on the net RF estimates (<10%, except in two cases where the difference compared to using 3D kernels is 20%) (see also Stjern et al. (2016) for a fuller discussion on this issue). We also estimate the Arctic average forcing and response to Arctic BC concentrations changes and briefly investigate the potential uncertainties in Arctic TS using this kernel-based approach due to influence from forcing exerted outside the region. The latter is done by using a kernel for the temperature response caused only by the local Arctic BC perturbation from Flanner (2013) (see Sect. 3.3)."*

P 10 L 307: AMAP models are shown to produce dramatically different seasonal variations in BC burdens in the Arctic (Mahmood et al., 2016). Currently available observational data sets are insufficient to validate seasonal variations in atmospheric BC burdens in the Arctic. Does this statement refer to near-surface concentrations?

The reviewer raises an important point. We have clarified that this statement, as well as the improvements found in several modeling studies, concerns near-surface concentrations, and that considerable inter-model variability remain. We have also included a reference to the Mahmood et al. (2016) paper, which is a very relevant study we missed in the original submission. The revised version reads:

*"While there has been considerable progress and several current models capture the seasonality in high-latitude surface concentrations better than previous generations [Breider et al., 2014; Browse et al., 2012; Liu et al., 2011; Sharma et al., 2013], discrepancies remain and there is considerable inter-model variability in simulated Arctic atmospheric BC burdens [Mahmood et al., 2016]."*

P 11 L 332: Please quantify. What does "somewhat high" mean?

We have added the following based on visual inspection of modeled concentrations in the studies in question:

*"In the present analysis, we find higher modeled concentrations than LB12, on the order of 5-15 ng g$^{-1}$ over large areas north of 70°N, likely owing to the updated emission inventory and shorter model time step for precipitation."*

Fig 3: Include a profile from the control simulation for comparison with concentration differences.
Added to supplementary as part of the response to major comment #2 above.

P 15 L 451: "...in order for models to reproduce the HIPPO data...". Clearly, these profiles cannot be "reproduced" in models but perhaps the general agreement with observations has been improved? Also, this depends on the metric that is used. Typically, BC concentrations near the surface are systematically too low in models and this bias tends to become more severe at reduced lifetimes.

We agree that the wording could be improved here and have rephrased:

*"This is in line with other recent studies showing that modeled BC vertical profiles agree better with HIPPO data, in particular at higher altitudes, when the global BC lifetime is reduced."*

P 18 L 544: Please provide RFari due to BC from all sources from the model. How does it compare to the IPCC estimate? How do the individual values of RFari in Fig. 6 compare to this value?
For comparison with the magnitude of absolute changes in RFari in the sensitivity experiments (Fig.6), we have added the RFari total BC estimated from the baseline simulation using the kernel method:

*"These changes are on the order of 25-50% of the RFari in the baseline simulation, estimated to be 0.35 W m$^{-2}$."*

However, this estimate does not account for the absorption enhancement due to BC aging and is relative to a case with no BC, not relative to pre-industrial, and should therefore not be compared to the IPCC value. We have therefore also added the following paragraph:

*"We emphasize that the RFari from the baseline simulation in the present study should not be compared with estimates of pre-industrial to present forcing by BC as it does not include the absorption enhancement due to BC aging and is given relative to a no-BC situation."*

P 20 L 606: For comparison, please provide the global mean surface temperature response that corresponds to RFari to BC from all sources. How do the temperature responses in Fig. 6 compare to this value?

Presumably, the reviewer means temperature response to ERFari here, corresponding to Fig.6? Given that we use a kernel-based approach to provide estimates of temperature response, and that this does not account for absorption enhancement, we have reservations about presenting a temperature response from the baseline simulation here in the conclusion section. However, to place the absolute changes in Fig.6 in context, we have estimated the baseline TS and added the following to the Section 3.3:

*"(…), resulting in a decrease of -14 to -25 mK compared to the baseline (13-22% of the TS in the baseline simulation estimated using the kernel-based approach)."*

References:

Allen, R. J., and W. Landuyt (2014), The vertical distribution of black car-
bon in CMIP5 models: Comparison to observations and the importance of convective transport,
J. Geophys. Res. Atmos., 119, 4808-4835, doi:10.1002/2014JD021595.

Mahmood, R., K. von Salzen, M. Flanner, M. Sand, J. Langner, H. Wang, and L. Huang (2016),
Seasonality of global and Arctic black carbon processes in the Arctic Monitoring and
Assessment Programme models, J. Geophys. Res. Atmos., 121, 7100-7116,
doi:10.1002/2016JD024849.